# Cryo-EM structures of the tubulin cofactors reveal the molecular basis of alpha/beta-tubulin biogenesis

Aryan Taheri, Zhoaqian Wang, Bharti Singal, Fei Guo & Jawdat Al-Bassam ✉

Microtubule polarity and dynamic polymerization arise from the self-association properties of the αβ-tubulin heterodimer. For decades, it has remained unclear how the tubulin cofactors TBCD, TBCE, TBCC, and the Arl2 GTPase mediate the biogenesis of αβ-tubulin from individual α- and β-tubulins. Here, we use cryo-electron microscopy to determine structures of tubulin cofactors bound to αβ-tubulin. TBCD, TBCE, and Arl2 form a heterotrimeric cage-like assembly, we term TBC-DEG, around the αβ-tubulin heterodimer. The TBC-DEG-αβ-tubulin structures show that TBC-DEG wraps around β-tubulin while TBCE extends along α-tubulin. The TBC-DEG/TBCC-αβ-tubulin structures reveal that TBCC forms multi-domain interactions with Arl2 and TBCD to engage the αβ-tubulin intradimer-interface, promoting TBCE rotation while TBCD holds β-tubulin. TBCC engages the GTP-bound Arl2, multiple sites of TBCD, and the native αβ-tubulin intradimer interface near the α-tubulin N-site GTP. Together, these structures uncover transition states for αβ-tubulin biogenesis and degradation, suggesting a vise-like, GTP-hydrolysis-dependent mechanism in which TBCC binding to TBC-DEG modulates αβ-tubulin inter-faces. Our studies provide structural evidence that tubulin cofactors act as enzymatic regulators that assemble the invariant αβ-tubulin architecture. By catalyzing α- and β-tubulin biogenesis and degradation, the TBC-DEG and TBCC assemblies regulate the polymerization competency of αβ-tubulin for microtubule formation.

Microtubules (MTs) are polarized cytoskeletal polymers that generate forces through dynamic polymerization at their ends[1–3], form stable tracks for intracellular cargo transport[4], and compose the rigid cores of cilia and flagella[5]. MTs are assembled by head-to-tail polymerized αβ-tubulin heterodimers into linear protofilaments, and lateral inter-actions between adjacent α- and β-tubulins stabilize the hollow, tube-like MT structures. The invariant organization of αβ-tubulin, with β-tubulin bound on top of α-tubulin, is fundamentally conserved across eukaryotes. Polymerization of αβ-tubulins promotes GTP-hydrolysis at the β-tubulin exchangeable site (E-site)[1,3,6]. A second non-exchangeable

(N-site) GTP on α-tubulin sandwiched between α and β-tubulins sta-bilizes the αβ-tubulin heterodimeric organization[3,7].

Eukaryotes have evolved specialized expression and folding mechanisms to maintain the high cellular concentration of α- and β-tubulin required for MT assembly. Translation of α- and β-tubulin mRNA is regulated by TTC5 and SCAPERs in response to intracellular αβ-tubulin concentration, while prefoldin delivers nascent α- and β-tubulin polypeptides to the CCT chaperonin complexes for ATP-dependent folding[8–11]. However, these pathways do not distinguish between α- and β-tubulin subunits. A conserved family of five tubulin

Molecular Cellular Biology Department, University of California, Davis, CA, USA. ²Present address: Molecular Cell Biology Department, University of California, Berkeley, CA, USA. ³Present address: Biochemistry Department, University of Washington, Seattle, WA, USA. ⁴Present address: Stanford Cryo-EM microscopy Center, Stanford University, Palo Alto, CA, USA. ✉e-mail: jmalbassam@ucdavis.edu

cofactors (TBCA, TBCB, TBCC, TBCE, TBCD, TBCE) assembles newly folded α- and β-tubulins into αβ-tubulin heterodimers through a GTP-hydrolysis-dependent process[8,9]. Orthologs of these cofactors regulate soluble αβ-tubulin concentration in across diverse eukaryotes and serve as the master regulators of MT polymerization[10–15]. TBCA and TBCB function in early assembly steps, whereas TBCC, TBCD, and TBCE perform central roles in αβ-tubulin biogenesis and are regulated by the conserved GTPase Arl2[13,14,16–20].

TBCC, TBCE, TBCD, and Arl2 are highly conserved across eukaryotes[9]. TBCD is composed entirely of α-helical HEAT repeats. TBCE contains an N-terminal Cap-Gly, central leucine-rich repeats (LRR), and C-terminal ubiquitin-like (ubq) domains. Arl2 consists of an N-terminal helix (Nh) and an Arf-like GTPase domain. TBCC consists of an N-terminal-helical bundle, a central disordered linker, and a C-terminal β-helix domain[8,9]. Complete inactivation of TBCs is lethal in many organisms due to loss of dynamic MTs[10,13,21,22]. Mutations in human TBCD, TBCE, and Arl2 cause developmental and neurological disorders[23,24]. TBCD mutations are linked to infantile neurodegenerative encephalopathy and corpus callosum hypoplasia[16,17,25,26]. TBCE N-terminal mutations are linked to hypoparathyroidism facial dysmorphism (Kenny-Caffey syndrome), while mutations of the TBCE C-terminus are linked to Giant Axonal motor neuropathy[18–20]. Mutations in Arl2 are linked to microcornea, rod-cone dystrophy, cataract, and posterior staphyloma (MCRS) syndrome[24]. Many of these disorders share impaired MT function during development.

A multi-subunit assembly of TBCD, TBCE, and TBCC has been proposed to promote αβ-tubulin biogenesis as a dynamic super-complex[12,21,22,27]. We previously demonstrated that Arl2 is the missing GTPase responsible for the GTP hydrolysis step of αβ-tubulin biogenesis[28]. The non-hydrolyzable GTP analog, GTPγS, prevents both αβ-tubulin release from TBC-DEG and TBCC dissociation[27]. Despite these advances, the structural basis of TBCD, TBCE, TBCC, and Arl2 organization and how they collectively mediate αβ-tubulin biogenesis and degradation pathways remains unknown.

Here, we present cryo-EM structures of the TBC-DEG-αβ-tubulin (binary) and TBC-DEG/TBCC-αβ-tubulin (ternary) assemblies, revealing that TBC-DEG forms an intricate cage-like assembly around the αβ-tubulin dimer, tightly encircling β-tubulin and binding along α-tubulin. TBCC recognizes the αβ-tubulin organization by bridging Arl2-GTP and the tubulin intradimer interface through long-range, multi-domain interactions. TBCC activates Arl2 GTP hydrolysis while simultaneously engaging the αβ-tubulin configuration. Our structures identify molecular transitions within the TBC-DEG-αβ-tubulin complex driven by TBCC binding and provide a mechanistic framework for understanding how αβ-tubulins are assembled, degraded, and recycled. These findings lead to a structural model potentially describing how TBCC-activated Arl2 GTP hydrolysis drives αβ-tubulin biogenesis.

## Results

### Cryo-EM structures of TBC-DEG-αβ-tubulin

We purified recombinant yeast TBC-DEG and reconstituted it with αβ-tubulin to form 1:1 TBC-DEG-αβ-tubulin (binary) assemblies, following our previous reconstitution strategy[28] (Fig. 1A and Supplementary Fig. 1). Initial cryo-EM sample preparation and imaging showed aggregation, consistent with self-association of bound αβ-tubulin. To improve sample homogeneity, we used a designer ankyrin repeat protein (DARPin) variant, ΔN-DARPin, which binds the β-tubulin polymerization interface and prevents αβ-tubulin self-association in the TBC-DEG-αβ-tubulin assemblies[29]. ΔN-DARPin binding did not alter the αβ-tubulin binding stoichiometry to TBC-DEG (Supplementary Fig. 1). Single-particle analysis of TBC-DEG-αβ-tubulin-ΔN-DARPin revealed homogeneous cone-shaped particles with clear secondary-structure features (Supplementary Fig. 2).

We determined cryo-EM structures of two states of the TBC-DEG-αβ-tubulin assemblies (state 1 and state 2) using a single-particle

reconstruction pipeline (Table 1 and Supplementary Figs. 2–4). The analysis yielded a consensus core TBC-DEG-αβ-tubulin structure at 3.6 Å global resolution and two lower-resolution states (~7 Å) for the arm extension representing TBCE region. The two TBCE states show domain reorganization and differ by a ~30° rotation around α-tubulin (Fig. 1E and Supplementary Figs. 3 and 4)[30]. We modeled the core TBC-DEG-αβ-tubulin assembly and the two conformations of the TBCE lever-arm extension (Table 1 and Supplementary Fig. 4).

AlphaFold3-predicted models of the TBC-DEG-αβ-tubulin assembly show a subunit organization and αβ-tubulin binding mode that closely match the experimentally determined structure (Fig. 1C, D and Supplementary Fig. 7)[31]. In the prediction, the αβ-tubulin intradimer interface exhibits a greater deformation relative to soluble αβ-tubulin than in the experimentally determined model.

### Organization of TBC-DEG-αβ-tubulin

The cryo-EM structures show that TBCD, TBCE, and Arl2 form a cage-like assembly around a single αβ-tubulin heterodimer (Fig. 1B, C). TBCD forms a ~100 Å ring composed of twenty-six α-helical HEAT repeats (H1–H26). TBCE is ~100 Å long and comprises an N-terminal Cap-Gly domain, an LRR domain with eight LRR repeats (LRR1–8), a three-helix bundle (3HB), and a C-terminal Ubq domain (Fig. 1A–C).

The N-terminal region of TBCD (termed the turret) binds and wraps around the protruding Arl2 Nh. The Arl2 GTPase domain sits above the TBCD turret and β-tubulin. The C-terminal region of TBCD (termed the spiral) contacts β-tubulin, and the full length of TBCD nearly encircles β-tubulin. The TBCE Ubq domain is anchored on top of the TBCD spiral, while the TBCE LRR domain lies beneath the spiral and alongside αβ-tubulin (Fig. 1B, C and Supplementary Movie 1). Alpha-Fold3 models closely match the experimentally determined TBCD and TBCE domain folds, TBC-DEG subunit interfaces, and TBCD–β-tubulin contacts (Fig. 1D and Supplementary Fig. 7).

TBCE adopts two distinct conformations of its Cap-Gly and LRR domains in the two cryo-EM states (Fig. 1E and Supplementary Fig. 4F, G). In state 1, the 3HB and LRR are oriented approximately vertically, and the Cap-Gly domain lies horizontally alongside the LRR (Fig. 1E and Supplementary Fig. 4F, G). In this conformation, TBCE is positioned away from the side of α-tubulin (Fig. 1B, C, Supplementary Movie 1, and Supplementary Fig. 4F, G). The TBCE Cap-Gly domain is close to the location of the α-tubulin C-terminal tail, but the resolution in this region is insufficient to model the tail or a specific Cap-Gly interaction. In state 2, the TBCE LRR domain is rotated by ~30° clockwise relative to state 1, via a hinge-like motion in the 3HB along the TBCD spiral (Fig. 1E, Supplementary Movie 1, and Supplementary Fig. 4F, G). In this conformation, the TBCE LRR runs along the lower side of α-tubulin and spans across the αβ intradimer region, while density for the Cap-Gly domain is not observed.

Although the experimentally determined and AlphaFold3-predicted TBC-DEG–αβ-tubulin models are very similar overall, the prediction differs in the conformation of the Arl2 GTPase and the orientation of the TBCE LRR–CapGly arm (Supplementary Fig. 7D). The AlphaFold3 TBC-DEG–bound αβ-tubulin heterodimer also shows a larger deviation from the conformation of soluble αβ-tubulin than does the experimental cryo-EM model (Supplementary Fig. 7E).

### TBCD is the scaffold of the TBC-DEG assembly

TBCD forms extensive contacts with Arl2, TBCE, and αβ-tubulin. The TBCD turret forms a ~60-Å-diameter solenoid through H1–H13 (Fig. 1F), and the TBCD spiral forms a crescent around β-tubulin via H14–H26 (Fig. 1C–F). The turret region encircles the Arl2 Nh and contacts the base of the Arl2 GTPase domain (Fig. 2A–D and Supplementary Movie 2). The Arl2 Nh is surrounded by the B-helices of H1, H2, H7, H8, and H9 (Fig. 2A, B, Supplementary Fig. 5A, and Supplementary Movie 2). Conserved residues in the intra-HEAT turns of TBCD H1, H2, H3, H5, H6, H8, and H10 contact residues on the lower surface of the Arl2 GTPase via ionic

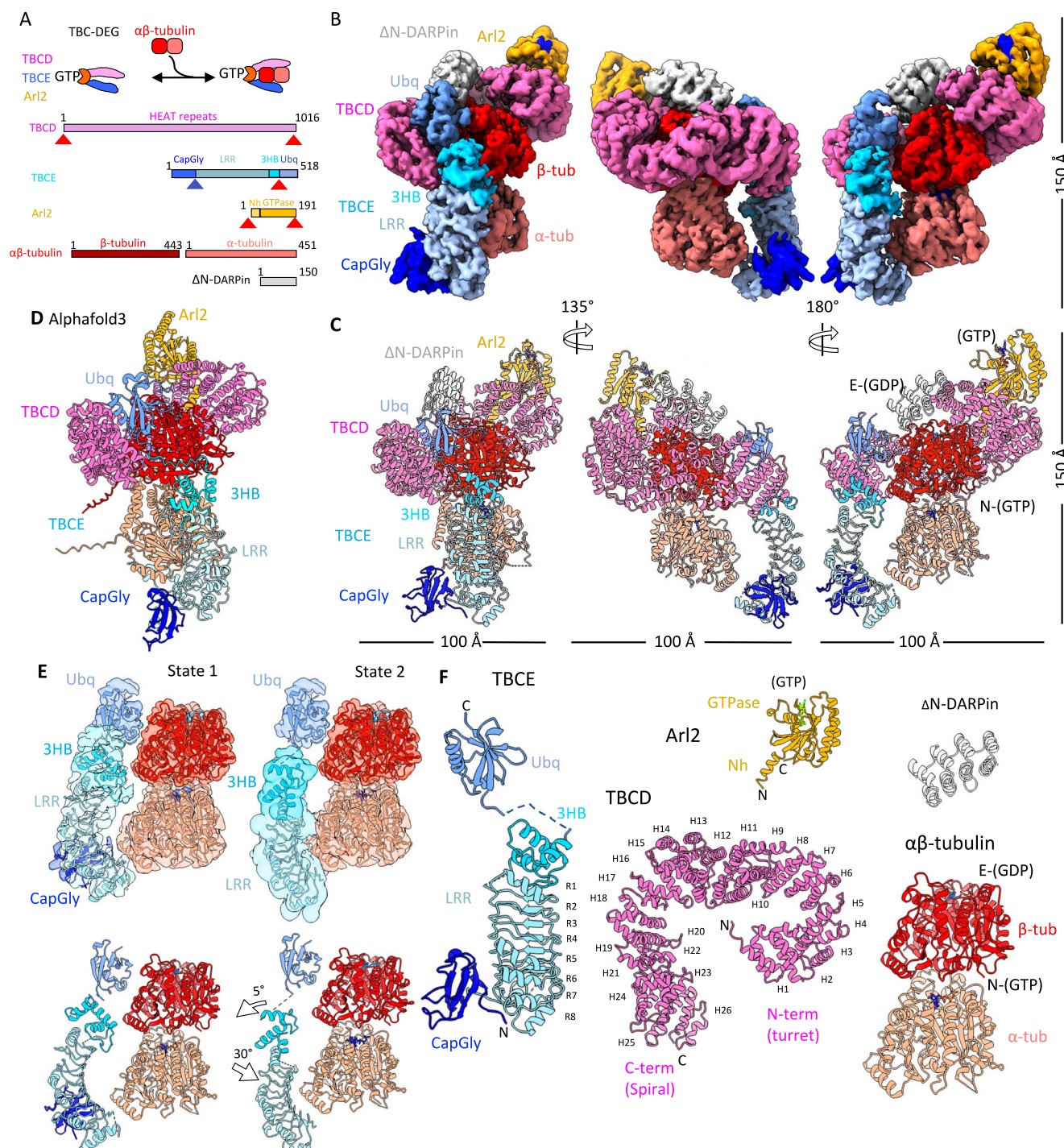

**Fig. 1 | Cryo-EM structures of TBC-DEG-αβ-tubulin (binary) assembly. A** Top, general organization of TBC-DEG in binding αβ-tubulin. Bottom, domains of subunits in the complex: TBCD (pink), second, TBCE (blue), Arl2 (orange), and DARPin (Grey). **B** Three views of the segmented cryo-EM reconstruction of TBC-DEG-αβ-tubulin-ΔN-DARPin, following the color scheme in (**A**). **C** Three views of cryo-EM atomic models of TBC-DEG-αβ-tubulin-ΔN-DARPin shown in the same orientations as in (**B**). **D** AlphaFold3 model for TBC-DEG-αβ-tubulin assembly shown in the same orientation as in the left panel of (**C**) and is described in more detail in Supplementary Fig. 7. **E** Two states of TBCE in binding to α-tubulin with the left panels showing State 1, right panels showing State 2. Top, cryo-EM segments of TBCE domains binding αβ-tubulin, Bottom models for TBCE binding αβ-tubulin. **F** TBC-DEG-αβ-tubulin subunit models. On the left, TBCE state 1 model is shown with annotated rotations and translations in the 3HB (3-helix bundle), LRR (leucine-rich Repeat), and CapGly domains. Second from the left bottom, a top view of TBCD with its twenty-six HEAT repeats, N-term (TBCD turret), and C-term (TBCD Spiral) domains. Second from the left top, Arl2 model (orange) is shown with its N-helix (N-h) and GTPase (GTPase) domains. Third at top left, DARPin model (Grey). Third from left bottom, αβ-tubulin model (tomato/red).

and hydrophobic interactions (Fig. 2C, D and Supplementary Figs. 5B, 20D, E, and 22D, E, G). Conserved TBCD Lys and Arg residues potentially contact conserved Arl2 Asp, Gln, and Glu residues (Supplementary Figs. 5, 17, 20D, E, G, and 22D, E), and conserved Leu, Phe and Trp

residues potentially contribute to hydrophobic packing (Fig. 2C, D and Supplementary Figs. 5, 17, 20D, E, G, and 22I, J).

The TBCD spiral binds the TBCE Ubq domain via the B-helices of TBCD repeats H18, H20, H23, and H25 (Fig. 2A–D, Supplementary

**Table 1 | Cryo-EM Data Collection, Single particle processing, refinement, model building, and validation statistics**

| Parameters | TBC-DEG -αβ-tubulin core | TBC-DEG-αβ-tubulin state 1 | TBC-DEG-αβ-tubulin state 2 | TBC-DEG-TBCC-αβ-tubulin core 1 | TBC-DEG-TBCC-αβ-tubulin TBCE 1 | TBC-DEG-TBCC-αβ-tubulin core 2 | TBC-DEG-TBCC-αβ-tubulin TBCE 2 | TBC-DEG-TBCC-αβ-tubulin TBCC |
|---|---|---|---|---|---|---|---|---|
| Microscope | Thermofisher Krios | | | Thermofisher Glacios | | | | |
| Detector | Gatan K2 | | | Gatan K3 | | | | |
| Magnification | 90000× | | | 45000× | | | | |
| Voltage (kV) | 300 | | | 200 | | | | |
| Electron exposure (e Å$^{-2}$) | 50 | | | 60 | | | | |
| Defocus range (µm) | −0.9 to −2.1 | | | −0.7 to −1.7 | | | | |
| Pixel size (Å/pix) | 0.662 | | | 0.44 | | | | |
| Symmetry imposed | *C1* | *C1* | *C1* | *C1* | *C1* | *C1* | *C1* | *C1* |
| No. of final particle images | 275,487 | 142,793 | 132,694 | 102987 | 177422 | 87013 | 370917 | 88829 |
| Resolution 0.143 FSC threshold (Å) | 3.6 | 3.7 | 3.7 | 3.7 | 3.6 | 3.8 | 3.5 | 3.7 |
| Initial Model | – | Alphafold3 | Alphafold3 | Alphafold3 | Alphafold3 | Alphafold3 | Alphafold3 | Alphafold3 |
| Model resolution FSC threshold | – | 0.5 | 0.5 | 0.143 | 0.143 | 0.143 | 0.143 | 0.143 |
| EMDB ID | EMD-70497 | EMD-70499 | EMD-70498 | EMD-70504 | EMD-70506 | EMD-70516 | EMD-70518 | EMD-70505 |
| Composite EMDB ID | | EMD-47949 | EMD-47954 | EMD-47947 | | EMD-47948 | | |
| Model refinement PDB ID | – | 9EDT | 9EEB | 9EDR | | 9EDS | | |
| Chains | – | 6 | 6 | 7 | | 7 | | |
| Atoms | – | 13115 | 12555 | 40263 | | 39102 | | |
| Residues | – | 2622 | 2510 | 2850 | | 2818 | | |
| Ligands | – | 3 | 3 | 3 | | 3 | | |
| Model Resolution | – | 4.4 | 4.1 | 6.5 | | 4.2 | | |
| Model Sharpening B factor | – | 115 | 75 | 95 | | 80 | | |
| **Validation** | | | | | | | | |
| MolProbity Score | – | 1.45 | 1.2 | 1.5 | | 1.78 | | |
| Clash Score | – | 4 | 2.08 | 7.46 | | 11.01 | | |
| Rotamer Outliers (%) | – | 0% | 0% | 0.04% | | 0.59% | | |
| Ramachandran (favored) (%) | – | 96.4% | 96.5% | 97.59% | | 96.53% | | |
| Ramachandran (allowed) (%) | -- | 3.4% | 3.4% | 2.38% | | 3.4% | | |
| Ramachandran (disallowed) (%) | – | 0.2% | 0.1% | 0.04% | | 0.07% | | |

Fig. 5A, and Supplementary Movie 2). This interface is dominated by a conserved hydrophobic interface between TBCD and TBCE Leu, Val, Phe, Trp, and Tyr residues (Fig. 2C, D and Supplementary Figs. 5A, 16–18, 20C, and 22C, H). TBCD H26 contacts the TBCE 3HB, and this region is stabilized by the adjacent Ubq–spiral interface (Figs. 1E and 2E, F and Supplementary Fig. 18). Both TBCD H26 and the TBCE 3HB are conserved across orthologs (Supplementary Fig. 20D).

In the TBC-DEG–αβ-tubulin structures, the N- and C-termini of TBCD, Arl2, and TBCE are either buried or participate directly in subunit interfaces with each other or with α- and β-tubulins (Supplementary Figs. 5, 6, and S20–S22). Consistent with this, truncation or fluorescent-protein fusions at these termini disrupt assembly reconstitutions[28].

**TBC-DEG encases a single αβ-tubulin heterodimer**
The TBC-DEG cage forms multiple interfaces with both α- and β-tubulin. The binding sites overlap with surfaces used for αβ-tubulin contacts in polymerized MTs and with sites recognized by many

MT-associated proteins (Fig. 2G–I). TBCD encircles β-tubulin and binds at three regions involving β-tubulin H2, H3, H4, H11, H12, and part of its unstructured C-terminal tail (sites I–III; Fig. 2G–I and Supplementary Fig. 6D–F). The extreme N-terminus of the Arl2 Nh lies near the GDP-occupied β-tubulin E-site (site II; Fig. 2G–I and Supplementary Figs. 6E and 20J). The TBCE Ubq domain, positioned by the TBCD spiral, forms a fourth binding site on β-tubulin (site IV; Fig. 2G–I and Supplementary Fig. 6G) near the β-tubulin H6/H7 helices and the M-loop. Together, these interfaces contact most of the β-tubulin polymerization surfaces (Fig. 2D, E and Supplementary Fig. 6), and the intervening TBCD regions form a nearly continuous ring surrounding, but without binding to β-tubulin (Fig. 2D, E and Supplementary Figs. 20–22).

The TBCE LRR–CapGly region extends along α-tubulin and covers additional lateral polymerization surfaces, including the H10 helix near the α-tubulin M-loop (Supplementary Fig. 4F, G). The TBCE LRR–CapGly arm runs longitudinally along αβ-tubulin and occupies two positions (Fig. 1C). In state 2 it contacts the lateral α-tubulin

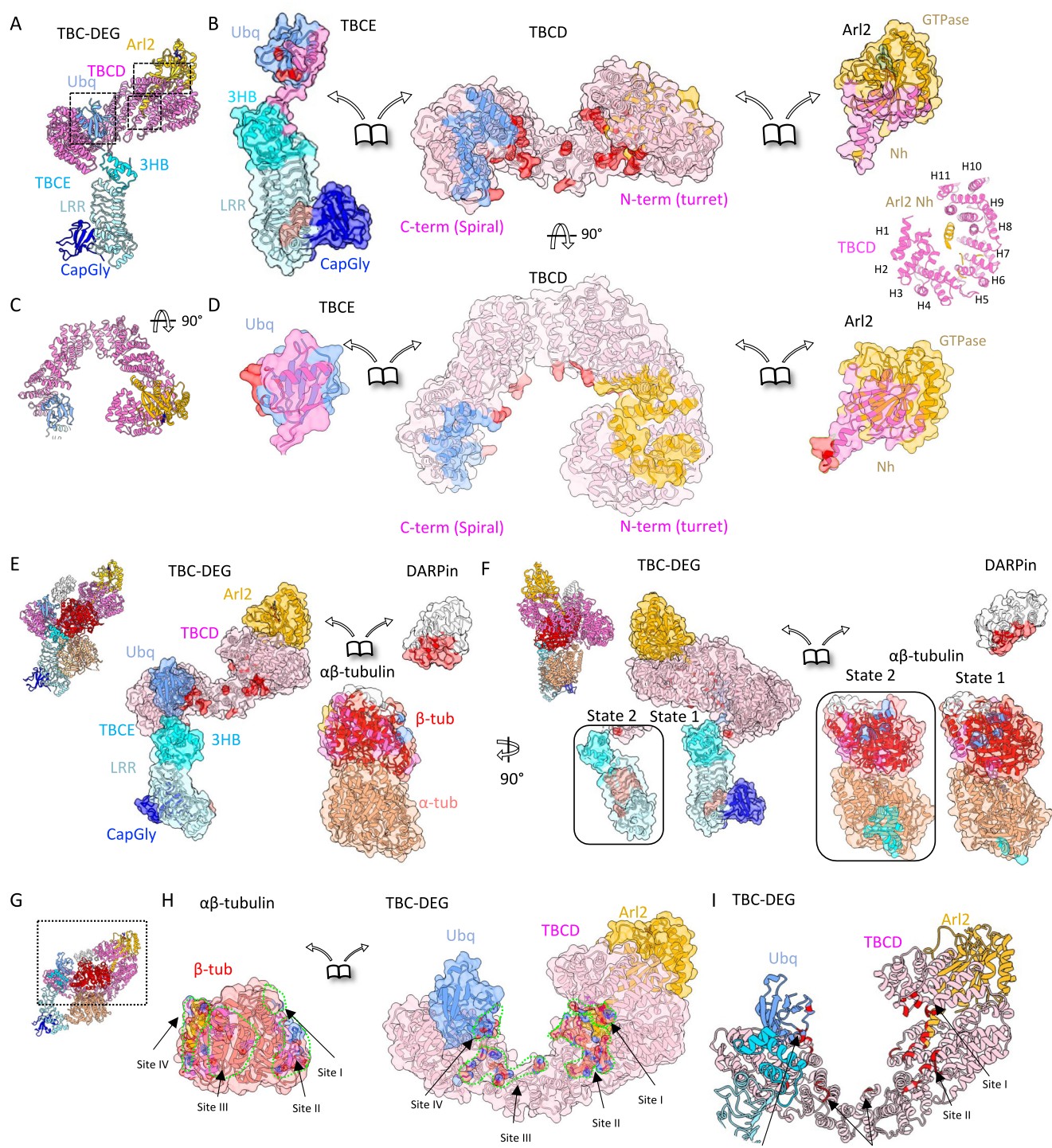

**Fig. 2 | The TBC-DEG cage-like assembly organization and interfaces with αβ-tubulin. A** Model for TBC-DEG assembly without αβ-tubulin. Boxes mark the interacting interfaces shown in (**B**, **C**). **B** Disassembled view of the TBC-DEG. TBCD is in the center, TBCE on the left, and Arl2 on the right. Interaction interfaces are colored by the footprints of the interacting subunits (TBCD: pink, TBCE Ubq: cyan, TBCE LRR: Sky blue, Arl2: orange, β-tubulin: red). At the bottom left, a close-up view of the Arl2 N-h interface with TBCD turret HEAT repeats is displayed. **C** 90° rotated model for overall TBC-DEG assembly without αβ-tubulin. **D** Disassembled view of TBC-DEG with subunit surface footprints colored by the interacting subunits and marking charged residues colored (red: negative; blue: positive). Details are shown in Supplementary Figs. 5 and S6. **E** TBC-DEG interactions with αβ-tubulin. Left, inset side views of the TBC-DEG-αβ-tubulin structure. Right panels, a disassembled view of TBC-DEG on the left, and αβ-tubulin on the bottom right and DARPin, top right.

The interaction interfaces are colored the same as (**B**). **F** 90° rotated inset view of TBC-DEG-αβ-tubulin structure shown in (**E**) left panel. Right panels, 90° rotated view of the disassembled view as shown in left panels in (**E**). Left box inserts below show the TBCE state 2 and its interaction surface with α-tubulin. **G** Side view of the TBC-DEG-αβ-tubulin structure highlighting the TBCD-β-tubulin interface. **H** TBC-DEG interacts with β-tubulin via four binding zones (marked zone I–IV). Left panel, Side view of TBC-DEG-β-tubulin with the region of focus boxed. The TBC-DEG (subunit colored surface) encircles β-tubulin (red ribbon). Right panels, a disassembled view of the left panel showing β-tubulin (left) TBC-DEG surface (right) with its interacting surface β-tubulin colored based on charge and sites marked with green outlines. β-tubulin binding surface colored by its interacting subunits. **I** A ribbon view of the TBC-DEG assembly with β-tubulin binding loops colored in red. The interaction interfaces (marked zone I-zone IV).

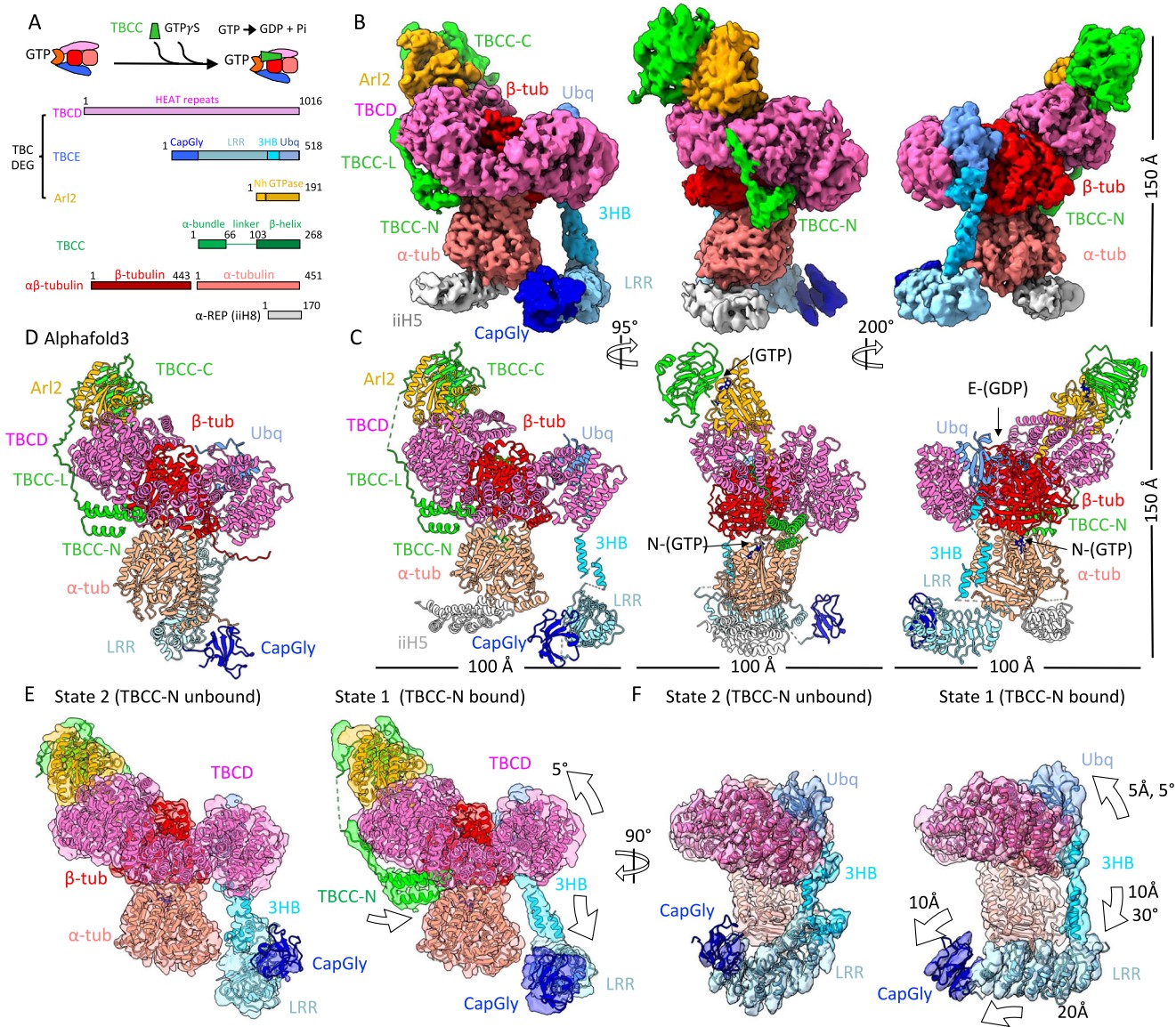

**Fig. 3 | Cryo-EM structures of the TBC-DEG/TBCC-αβ-tubulin (ternary) assembly reveal the catalytic transitions induced by TBCC binding. A** Top, general organization of TBC-DEG-αβ-tubulin in binding TBCC and GTPγS. Bottom, Linear domains of subunits in the complex: TBCD (pink), second, TBCE (blue), Arl2 (orange), TBCC (green), and iiH5 (Grey). **B** 3.6-Å Cryo-EM segmented TBC-DEG/TBCC-αβ-tubulin-iiH5 assembly shown in different views. Different domains are colored based on the scheme shown in Fig. 1A and as labeled. **C** Atomic models of TBC-DEG/TBCC-αβ-tubulin-iiH5 are shown in identical views to (**B**). Subunits, domains, and elements are colored and labeled as shown in (**A**, **B**). **D** Predicted TBC-DEG/TBCC-αβ-tubulin AlphaFold3 model shown in the same orientation as in left panel in (**C**) and is described in more detail in Supplementary Fig. 16. **E** Side views for two cryo-EM states: left, state 2; right, state 1. Cryo-EM reconstruction segments (transparent) with atomic models (ribbon) representing two distinct TBC-DEG/TBCC-αβ-tubulin subunits. Left, State 2 shows TBCC-N (green) not bound to the TBC-DEG-αβ-tubulin core. Right, State 1 shows TBCC-N (green) is bound to the TBC-DEG-αβ-tubulin core. **F** 90° rotated slice views compared to (**E**), of State 2 (left) and State 1 (right), showing TBCE LRR-CapGly arm rotation states and TBCD transitions marked by arrows.

surface via the LRR, whereas in state 1 it is positioned away from α-tubulin (Fig. 2E, F and Supplementary Fig. 4F, G). Comparison with soluble αβ-tubulin shows a modest clockwise twist of the heterodimer and a ~2-Å shift of α-tubulin in the TBC-DEG−bound state (described in detail below). In state 1, the TBCE CapGly region lies near the unstructured α-tubulin C-terminal tail, but the tail itself is not resolved in the density (Figs. 1 and 2C, D and Supplementary Movie 2).

**Cryo-EM structures of TBC-DEG/TBCC-αβ-tubulin**
We reconstituted recombinant TBC-DEG with TBCC and αβ-tubulin to form TBC-DEG/TBCC–αβ-tubulin (ternary) assemblies for single-particle cryo-EM[28]. TBCC binds TBC-DEG only when αβ-tubulin and GTP (or analogs) are present[28]. High-affinity TBCC binding was

achieved using an Arl2-Q73L TBC-DEG variant, or by adding GTPγS[28] (Fig. 3A and Supplementary Fig. 8A, B). Stoichiometric TBC-DEG−Arl2-Q73L, αβ-tubulin, and TBCC complexes were isolated by size-exclusion chromatography in the presence of 5 mM GTPγS (Supplementary Fig. 8C). Initial cryo-EM imaging showed aggregation of ternary assemblies. The addition of ΔN-DARPin reduced aggregation but also reduced TBCC occupancy, so we used the α-rep iiH5, which binds the minus end of α-tubulin[32]. iiH5 did not alter TBCC stoichiometry and yielded homogeneous particles (Supplementary Figs. 8 and 9).

Single-particle analysis of TBC-DEG/TBCC–αβ-tubulin ("Methods"; Supplementary Figs. 9–11) produced a 4.0-Å consensus core structure. An arm shaped density dissociated from the core was attributed to TBCE LRR−CapGly based on connectivity and domain

size (Supplementary Fig. 9), and this conformation differs from the bound TBCE LRR–CapGly state seen in the binary assemblies (Supplementary Fig. 6). Several regions showed blurred density consistent with conformational heterogeneity (Supplementary Fig. 9). To better characterize these regions, we performed 3D variability analysis (3DVA) with four components in CryoSPARC[30] (Supplementary Figs. 9 and 10). The 3DVA reconstructions revealed conformations with TBCC N-terminal (TBCC-N) and C-terminal (TBCC-C) domains either present or absent at their binding sites (component 1; Supplementary Fig. 10A), two poses of the TBCD spiral/TBCE-Ubq region differing by ~5° (component 2; Supplementary Fig. 10B), two conformations of TBCE-LRR–CapGly associated with TBCC-N occupancy (component 3; Supplementary Fig. 10C), and three orientations of the TBCE LRR-CapGly arm (component 4; Supplementary Fig. 7D). Using these classes, we generated two composite maps representing two conformations of the ternary assemblies (Supplementary Fig. 12).

AlphaFold3 models of TBC-DEG/TBCC–αβ-tubulin match the experimentally determined subunit arrangement, TBCC multi-domain interactions, and TBCC–αβ-tubulin contacts (Fig. 3C, D and Supplementary Fig. 16). The predicted model shows a different orientation of the TBCE LRR–CapGly arm and a smaller deformation of the αβ-tubulin intradimer interface compared with the TBC-DEG–only predicted model (Supplementary Fig. 16D, E).

### TBCC binds Arl2, TBCD, and engages αβ-tubulin intradimer interface near N-site

The two composite TBC-DEG/TBCC–αβ-tubulin structures show how TBCC binds to the TBC-DEG–αβ-tubulin core (Fig. 3B, C, Supplementary Figs. 9–13, and Supplementary Movie 3). TBCC contains an N-terminal α-helical bundle (TBCC-N; residues 1–75), a linker (TBCC-L; residues 76–100), and a C-terminal β-helix domain (TBCC-C; residues 100–270) (Supplementary Fig. 15B, Fig. 3B, C, and Supplementary Movie 3). The two ternary structures differ in the presence or absence of TBCC-N density at a site between the central TBCD region and the αβ-tubulin intradimer interface. When TBCC-N density is present, the conformations of the TBCD spiral, TBCE, and αβ-tubulin differ from those in the TBCC-N-absent map (Fig. 3E, F). In both ternary structures, TBCC-C and TBCC-N correlate with TBCE LRR being displaced from α-tubulin and rotated relative to the binary assembly (Fig. 3E, F).

The TBCC-C β-helix contacts the Arl2 GTPase in a manner similar to the RP2–Arl3 complex (Figs. 3B–E and 4)[28]. TBCC-C, Arl2, and the TBCD turret form an interface comprised of conserved ionic and hydrophobic interactions (Fig. 4A–C and Supplementary Movie 4). Modeling the TBCC-C crystal structure into TBCC-C density, as well as comparison to the Alphafold3 model suggests that TBCC-C residues Arg186 and Gln184 likely contact the Arl2-bound nucleotide (Fig. 4C and Supplementary Figs. 19 and 20A, B, E)[28]. This Arg186 has been reported to be crucial for the Arl2 GTPase in the TBC-DEG GTP hydrolysis[28]. A conserved TBCC-C loop contacts the TBCD turret beneath the Arl2-GTpase; Deletion of the TBCC-C loop strongly decreases the Arl2 GTP hydrolysis within TBC-DEG in response to αβ-tubulin binding[28] (Figs. 3B and 4 and Supplementary Fig. 15C). TBCC-L segment runs along TBCD, and the conserved N-terminal portion of TBCC-L binds between TBCD repeats H11 and H12 (Fig. 4A, B, Supplementary Movie 4, and Supplementary Figs. 15 and 19); the central region of TBCC-L is not resolved and presumed to be disordered.

The TBCC-N three-helix bundle binds beneath the central TBCD region (Fig. 4A, B and Supplementary Movie 3). TBCC-N contacts the αβ-tubulin intradimer interface, primarily α-tubulin, and is positioned ~8 Å from the γ-phosphate of the α-tubulin N-site GTP (Figs. 3 and 4A, B and Supplementary Movies 3 and 4). TBCC-N contacts both α-tubulin and β-tubulin, binding their intradimer interface at α-tubulin H2 and H3 residue,s likely by acidic Glu and Asp residues and β-tubulin H6 Glu residues. Structural and sequence conservation suggests that the TBCC-N bundle is extended by a longer α-helix in metazoan species,

while it remains short in yeast orthologs (Supplementary Fig. 19C). The TBCC-N interactions bridge between wedging interactions against TBCD in TBC-DEG and the recognize the αβ-tubulin intradimer interface.

AlphaFold3 models of TBC-DEG/TBCC–αβ-tubulin recapitulate the subunit organization and TBCC multi-domain contacts seen in the cryo-EM structures (Supplementary Fig. 16A–C). As in the TBC-DEG–only predictions (Supplementary Fig. 7), the predicted TBCE LRR–CapGly orientation differs from that seen in the experimental maps (Supplementary Fig. 16D). The predicted ternary model shows the α-tubulin position closer to its conformation in soluble αβ-tubulin compared to the α-tubulin position in the TBC-DEG-αβ-tubulin predicted model (Supplementary Fig. 16E).

### TBCC catalyzes transitions in TBC-DEG-αβ-tubulin

Comparisons between the two binary and two ternary assembly states highlight conformational differences among TBC-DEG subunits and αβ-tubulin and the impacts of TBCC domains binding (Fig. 5A and Supplementary Figs. 4 and 10). The largest differences observed involve the TBCE LRR–CapGly arm. In state I (TBC-DEG–αβ-tubulin state 2), the TBCE LRR lies along the side of α-tubulin (Fig. 5A, I). In state II (TBC-DEG-αβ-tubulin state 1), TBCE is positioned away from α-tubulin and runs parallel to the αβ-tubulin dimer, with CapGly adjacent to the LRR (Fig. 5A, II). In state III (TBC-DEG/TBCC–αβ-tubulin state 2), density for TBCC-C is present at the Arl2 GTPase and TBCD turret (Fig. 5A, III and Supplementary Fig. 15D), and the TBCE LRR–CapGly arm is rotated downward and clockwise relative to state II (Supplementary Fig. 14A, B and Fig. 5A, III). In state IV (TBC-DEG/TBCC–αβ-tubulin state 1), TBCE LRR is rotated further clockwise and translated relative to state III, and the distal portion of the TBCD spiral and TBCE Ubq are shifted upward by ~5° and ~5 Å (Fig. 5A, IV and Supplementary Fig. 14A, B).

In these four states, TBCE LRR–CapGly arm undergoes a stepwise series of positional changes: from a conformation in which it contacts α-tubulin using its LLR interface (state I), to an unbound conformation aligned along the dimer (state II), and to two rotated conformations seen in the ternary complexes (Fig. 5A, III–IV). Between states II and III, α-tubulin exhibits a ~1.5-Å shift at the intradimer interface (Fig. 5B, C). TBCC binding induces the dissociation of TBCE LRR CapGly from α-tubulin (Fig. 5A, III-IV). The translation observed in α-tubulin in the TBC-DEG–αβ-tubulin state and its reduction in the ternary complex are also seen in the corresponding AlphaFold3 models (Fig. 5C, D and Supplementary Figs. 7E and 16E). Across the four structures, the TBC-DEG-αβ-tubulin binary assemblies show a larger deviation of the αβ-tubulin intradimer geometry from soluble αβ-tubulin than those observed in the TBC-DEG/TBCC–αβ-tubulin ternary assemblies (Fig. 5C, Supplementary Fig. 14C–E and Fig. 5D).

## Discussion

Using biochemical reconstitution, cryo-EM structures, and AlphaFold3 models, we reveal the organization of the conserved TBC-DEG assemblies, their extensive interaction interfaces with αβ-tubulin, and their TBCC-dependent structural transitions. Our studies lead to a model for TBC-DEG and TBCC activity in regulating αβ-tubulin states during biogenesis and degradation (Fig. 6). We show that TBCD, TBCE, and Arl2 form TBC-DEG cage-like assemblies that bind to individual αβ-tubulins (Figs. 1 and 2). TBCC binds, with its multiple domains, to these assemblies and promotes TBCE to undergo catalytically activated transitions that induce a 2-Å translation of α-tubulin while β-tubulin is tightly held in place by TBCD (Figs. 3 and 4). The conserved topology of αβ-tubulin requires energy for its assembly[9,28]. Our structural studies suggest that TBCC binding catalyzes the Arl2 GTP hydrolysis, activates these transitions within the αβ-tubulin bound TBC-DEG assembly. The TBC-DEG appears to overcome αβ-tubulin stability by altering the αβ-tubulin intradimer interface near the α-tubulin N-site

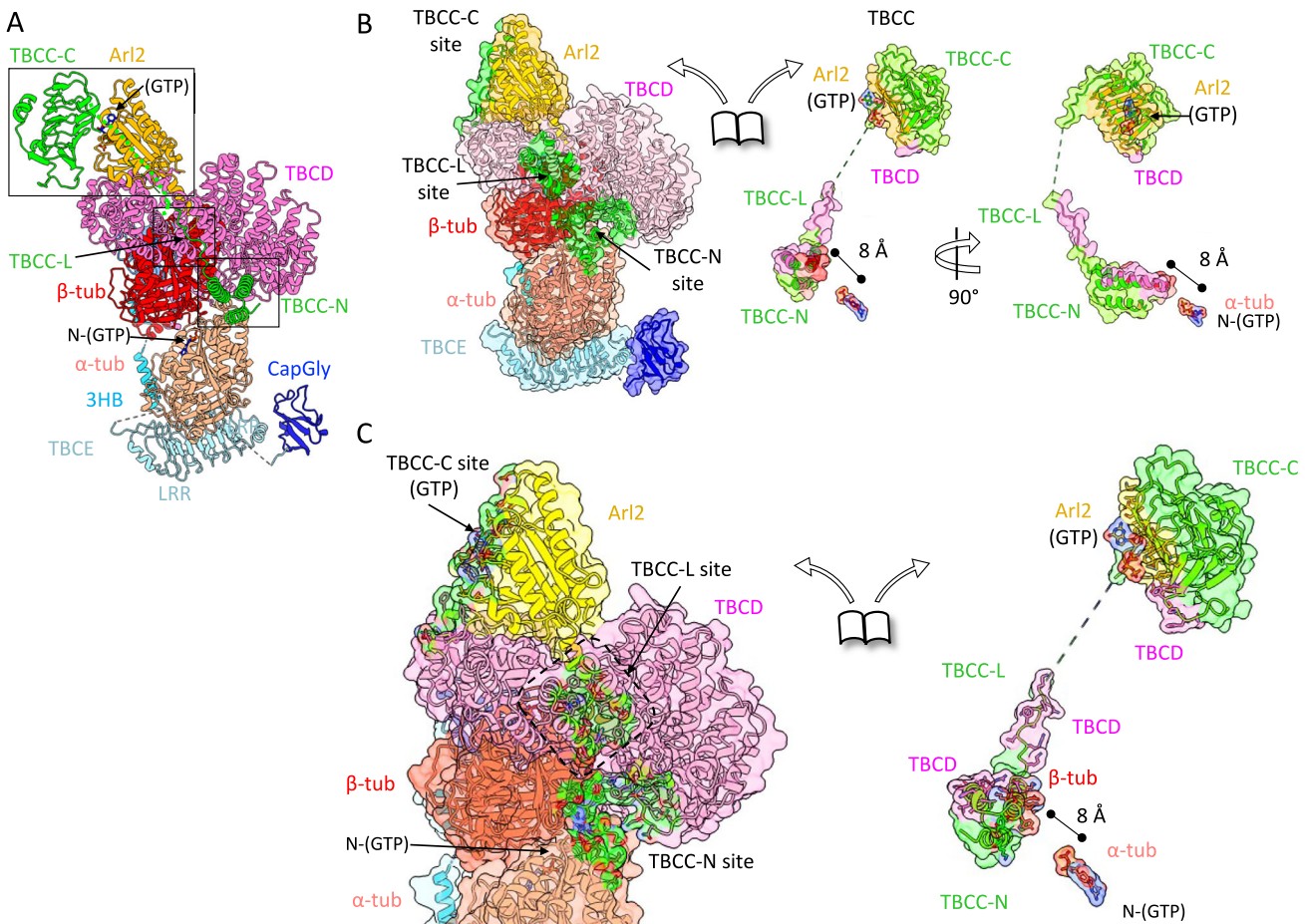

**Fig. 4 | The interfaces of TBCC with TBC-DEG-αβ-tubulin. A** Overall side view of TBC-DEG/TBCC-αβ-tubulin model highlighting the three interaction interfaces of TBCC-C, TBCC-L, and TBCC-N with Arl2, TBCD, and αβ-tubulin regions.
**B** Disassembled view showing the TBC-DEG-αβ-tubulin core (surface with ribbon, left panel) with the footprint of TBCC marked and TBCC-C, TBCC-L, and TBCC-N binding sites. Right, TBCC subunit (surface with ribbon) marked for its interaction interfaces with TBC-DEG-αβ-tubulin subunits. Interaction interfaces are colored by the footprints of the interacting subunits (TBCD: TBCE Ubq: cyan, TBCE LRR: Sky blue, Arl2: orange, β-tubulin: red). The Arl2 GTP binds TBCC-C directly and N-site GTP in α-tubulin is 8 Å away from TBCC-N domain tip. **C** Close up disassembled view of TBCC interaction interfaces colored to show charged residues. Disassembled view of the TBC-DEG/TBCC-αβ-tubulin model showing the three TBCC interaction interfaces with residues colored based on their charge (red, negative; blue positive): TBCC-C: Arl2-GTP and TBCD, TBCC-L: TBCD and TBCC-N: TBCD, α- and β-tubulin. Detailed interactions are shown in Supplementary Fig. 16.

GTP. TBCC binding and Arl2-GTP hydrolysis then promote transitions that restore a more native αβ-tubulin intradimer interface (Fig. 6). The close agreement between the cryo-EM structures and the AlphaFold3 models suggests that the high level of conservation of TBCD, TBCE, Arl2, TBCC, α-tubulin, and β-tubulin supports this organization and the conformational changes we observe. The AlphaFold3 models predict a larger deformation of αβ-tubulin than we observe experimentally, and this difference may reflect the lower catalytic potential of the yeast TBC-DEG used here, together with porcine αβ-tubulin (Fig. 5C, D).

Our model suggests that TBC-DEG binding to αβ-tubulin alters the α-tubulin configuration while tightly holding β-tubulin, and that TBCC binding then drives changes that restore a near native αβ-tubulin conformation (Figs. 5A and 6). The binding of TBCC-C to Arl2 upon GTP binding, followed by TBCC-L binding to TBCD, appears to position TBCC-N beneath TBCD, where it contacts the αβ-tubulin intradimer interface (Fig. 5A, IV). TBCC-N binding is associated with rotation of TBCE, mediated by an upward movement of the TBCD spiral and a downward rotation of the TBCE 3HB region (Fig. 5A, II–IV). This motion shifts the TBCE LRR-CapGly region clockwise and brings the CapGly domain near the α-tubulin C-terminal tail (Fig. 5A, IV). Although the CapGly-α-tubulin C-terminal tail interaction could not be modeled, its proximity suggests a potential functional role. These observations

support a model in which TBCC-activated Arl2 GTP hydrolysis provides the energy required for each αβ-tubulin assembly event. The TBC-DEG and TBCC cycle may also reset α-tubulin and β-tubulin by binding, altering, and reforming the configuration of soluble αβ-tubulin, a process sometimes described as tubulin recycling, which could be important for restoring the polymerization capacity of tubulin in vivo (Fig. 6). The requirement for Arl2 GTP hydrolysis is consistent with defects observed in Arl2 GTP- or GDP-locked mutants during interphase and mitosis. Although speculative, this model provides a possible explanation for how GTP hydrolysis by Arl2 contributes to αβ-tubulin formation and renewal.

A recent study published while this work was in review presents structures of the human tubulin cofactors and Arl2 in complex with α- and β-tubulin, leading to closely parallel discoveries on the organization of the TBC-DEG and multi-domain interactions of TBCC[33]. Their study reached similar conclusions about conformational transitions, with some differences concerning the conformational transitions of TBCE and the impact of TBC-DEG on α-tubulin conformation within αβ-tubulin. Notably, the TBC-DEG and TBCC bound α–tubulin states in their structures more closely resemble the conformations predicted in our Alphafold3 models (Supplementary Figs. 7 and 16) than those observed in our experimental data, likely due differing reaction

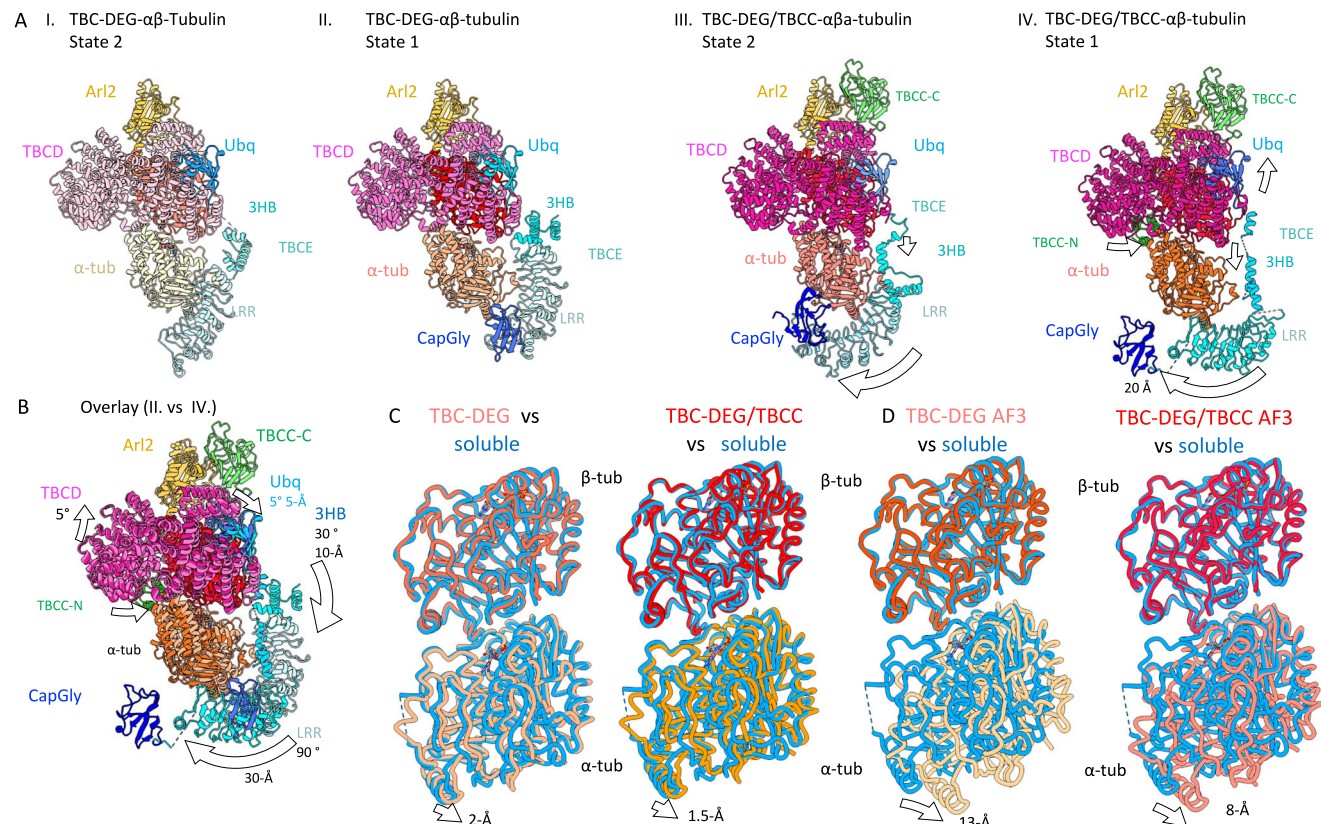

**Fig. 5 | Conformational transition of TBC-DEG and their impact on αβ-tubulin organization. A** I. TBC-DEG-αβ-tubulin State 2, II. TBC-DEG-αβ-tubulin State 1, III. TBC-DEG/TBCC-αβ-tubulin, state 2, IV. State IV (TBC-DEG/TBCC-αβ-tubulin, state 2). **B** An overlay of models, shown in II. vs IV. Subunits of TBC-DEG-αβ-tubulin state 1 are shown as in lighter colors, while TBC-DEG/TBCC-αβ-tubulin subunits are shown in darker colors. The conformational changes in α-tubulin, TBCD, TBCE domains are marked with arrows. **C** Left, Comparison of αβ-tubulin model in TBC-DEG-αβ-tubulin (light red) aligned to soluble αβ-tubulin (cyan). Right, the TBC-DEG/TBCC-αβ-tubulin (dark red) aligned to αβ-tubulin (cyan). In both β-tubulin is used as an alignment reference. **More details in** Supplementary Figs. 7, 14, and 16. **D** Left, Comparison of αβ-tubulin model in Alpahfold3 (AF3) of TBC-DEG-αβ-tubulin (light red) aligned to soluble αβ-tubulin (cyan). Right, the Alphafold3 (AF3) model of TBC-DEG/TBCC-αβ-tubulin (dark red) aligned to αβ-tubulin (cyan). In both β-tubulin is used as an alignment reference. **More details in** Supplementary Figs. 7, 14, and 16.

temperatures or usage of capping proteins. Together, the two studies provide complementary insights into the catalytic states of TBC-DEG during tubulin biogenesis and degradation.

## Tubulin cofactor assemblies are the GTP-dependent catalytic regulators of soluble αβ-tubulin

Eukaryotic cells maintain stoichiometric α- and β-tubulin expression, which is essential for MT function. Overexpression of β-tubulin is more toxic than α-tubulin, and excess β-tubulin leads to formation of insoluble aggregates, whereas excess α-tubulin does not. In both cases, MT functions are severely disrupted[34]. The extensive TBC-DEG binding surface surrounding β-tubulin observed in the structures suggests that TBC-DEG functions to isolate β-tubulin and prevent formation of β-tubulin aggregates until the αβ-tubulin heterodimer is assembled. The TBC-DEG structures presented here suggest that process of αβ-tubulin assembly and disassembly are catalyzed within this assembly. The regulatory activity of TBCC likely promotes the Arl2-GTP hydrolysis-dependent assembly of αβ-tubulin as evidenced by the near native organization of αβ-tubulin the TBC-DEG/TBCC-αβ-tubulin

The engaged or inaccessible N- and C-termini of TBCD, TBCE, Arl2, and TBCC in our structures provide an explanation for why these assemblies have been difficult to detect in vivo[21,22], despite extensive genetic and cell biological studies[35]. The structural data suggest that fusing fluorescent proteins to these termini likely disrupts TBC-DEG assembly or TBCC function[28], which may explain past difficulties in visualizing these complexes[36]. The cryo-EM structures also reveal that TBC-DEG binds across the α- and β-tubulin surfaces. This provides a

structural explanation for why large fluorescent tags at the tubulin C-termini and the β-tubulin N-terminus fail to complement tubulin gene deletions[37]. The exposed α-tubulin N-terminus, which is the only site where large tags can be tolerated, is consistent with its accessibility in our structures. Smaller epitope tags can be accommodated at either tubulin terminus, consistent with the space available within the TBC-DEG interfaces[37].

The shared functions of TBCD, TBCE, TBCC, and Arl2 in tubulin biogenesis and degradation align with observations that over-expression of TBCD or TBCE produces severe MT defects[36]. Over-expression likely disrupts their stoichiometric balance relative to TBCC, favoring a degradation-like pathway[35,36]. Our model suggests that the isolation of β-tubulin within TBC-DEG during both assembly and degradation cycles is important due to the intrinsic toxicity of excess β-tubulin. These findings imply that cellular stoichiometry between TBC-DEG and TBCC may be central to balancing biogenesis and degradation of αβ-tubulin. The requirement for Arl2 GTP hydrolysis further supports the idea that energy input is necessary to resolve the intrinsically stable αβ-tubulin interface and may enable a reset of tubulin to a conformation more compatible with MT polymerization.

## Methods
### Protein expression and purification

Recombinant TBC-DEG was purified as previously described[28]. Poly-cistronic expression construct were co-transformed into *Escherichia coli* SoluBL21 cells (AMSBIO). One construct encoded budding yeast *Saccharomyces cerevisiae* TBCD (*Cin1*) with an N-terminal 6 × His tag,

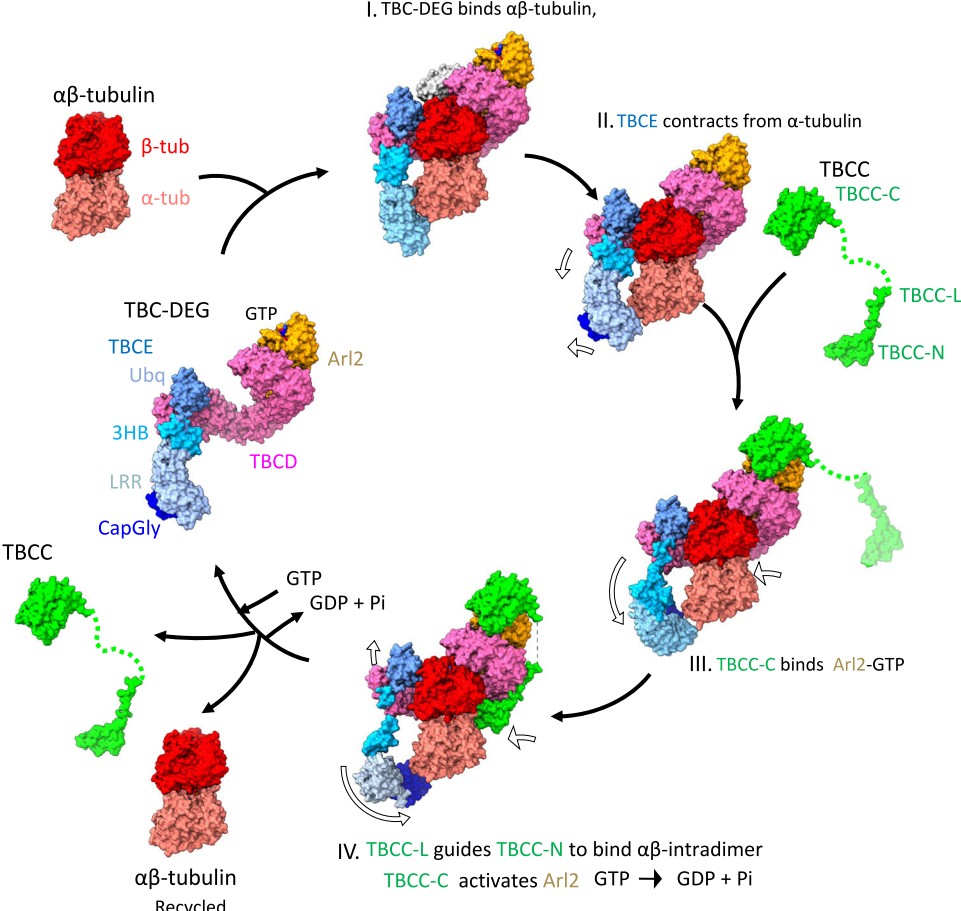

**Fig. 6 | Model of a catalytic cycle for TBC-DEG/TBCC in the biogenesis, degradation, and recycling of αβ-tubulin.** Left, TBC-DEG binds αβ-tubulin. I., TBC-DEG engages α-tubulin by its TBCE-LRR-CapGly and modifies αβ-intradimer interface. II. the TBCE-LRR-CapGly retracts from α-tubulin. III., TBCC-C binds to the Arl2-GTP, leading to changes in TBCE 3HB/TBCD spiral and TBCE pulls α-tubulin C-term tail. IV., TBCC-L guides TBCC-N to bind underneath TBCD and engages the αβ-tubulin interface, causing TBCE-LRR-CapGly to rotate, TBCC activates Arl2 GTP hydrolysis. TBC-DEG releases TBCC and αβ-tubulin and GTP and GDP are exchanged.

and the second encoded TBCE (*Pac2*) together with either wild-type or a GTP-locked mutant (Q73L) of Arl2 (*Cin4*) were co-transformed and transformants were selected with ampicillin and kanamycin. Cells were grown at 37 °C to an optical density 600 nm of 0.6 and induced with 0.5 mM isopropyl-β-D-thiogalactopyranoside (IPTG). Culture temperatures were shifted to 19 °C and grown overnight. Cells were pelleted, resuspended in lysis buffer (50 mM HEPES, 150 mM KCl, 1 mM MgCl2, 3 mM β-mercaptoethanol) supplemented with protease inhibitors (1 mM PMSF, 1 μg/mL leupeptin, 20 μg/mL benzamidine, 40 μg/mL aprotinin; Sigma Aldrich), and lysed using a microfluidizer. Lysates were clarified by centrifugation at 37,000 × *g* for 30 min at 4 °C. TBC-DEG assemblies was purified by Ni-IDA affinity chromatography (Macherey-Nagel), and bound protein was eluted with lysis buffer containing 200 mM imidazole. The eluted fractions were diluted to low-salt conditions (100 mM KCl, 50 mM HEPES, 1 mM MgCl2) and applied to a HiTrap SP FF cation-exchange column (Cytiva). Protein was eluted over five column volumes using a gradient to high-salt buffer (50 mM HEPES, 500 mM KCl, 1 mM MgCl2). Ion-exchange fractions containing TBC-DEG assemblies were concentrated using Amicon concentrators (Fisher Scientific) and further purified on a HiLoad 16/600 Superdex 200 column (Cytiva). Final peak fractions were concentrated, aliquoted, and flash-frozen in liquid nitrogen.

Recombinant and untagged budding yeast Saccharomyces cerevisiae budding yeast TBCC (*Cin2*) was expressed in SoluBL21 cells and purified using a previously published protocol[28]. A T7 plasmid containing untagged TBCC sequence was transformed in SoluBL21 cells

and selected with ampicillin. Cells were grown at 37 °C to an optical density 600 nm of 0.6 and induced with 0.5 mM IPTG. Culture temperatures were shifted to 19 °C and grown overnight. Cells were pelleted, resuspended in lysis buffer (50 mM PIPES, 100 mM KCl, 1 mM MgCl2, 3 mM β-mercaptoethanol pH 6.0) supplemented with protease inhibitors (1 mM PMSF, 1 μg/mL leupeptin, 20 μg/mL benzamidine, 40 μg/mL aprotinin (Sigma Aldrich), and lysed using a microfluidizer. Lysates were clarified by centrifugation at 37,000 × *g* for 30 min at 4 °C. TBCC was purified using ion exchange chromatography and eluted with 50 mM MES, 500 mM KCl, pH 6.0. TBCC-containing fractions were concentrated using an Amicon concentrator (Fisher Scientific). Soluble αβ-tubulin was purified using a temperature cycling procedure and GTP polymerization procotol in high PIPES buffer condition from porcine brain tissue using established procedure[38].

Codon-optimized constructs for ΔN-DARPin and α-Rep iH5 were synthesized based on published sequences[32,39]. Their coding sequence for ΔN-DARPin and α-Rep iH5 were cloned into T7 vectors containing either an N-terminal 6×His tag (ΔN-DARPin) or a C-terminal StrepII tag (iH5), respectively. Proteins were transformed into SoluBL21 cells and selected for transformants with Ampicillin. Transformed cells were grown at 37 °C and induced 0.5 mM IPTG at Optical Density 600 nm of 0.6, followed by incubation overnight at 19 °C. Cells were lysed using a microfluidizer in lysis buffer (50 mM HEPES, 300 mM KCl, pH 7.0, 3 mM β-mercaptoethanol) supplemented with protease inhibitors. ΔN-DARPin and iH5 were purified using Ni-IDA or StrepTactin XT affinity resins, respectively, and eluted with lysis buffer supplemented with

either 100 mM imidazole (ΔN-DARPin) or 50 mM biotin (iH5), respectively. Proteins were further purified by ion exchange on a HiTrap Q column (Cytiva) under low-salt conditions (50 mM HEPES, 100 mM KCl, 3 mM β-mercaptoethanol) and eluted across five column volumes using a gradient to high-salt buffer (50 mM HEPES, 1 M KCl, 3 mM β-mercaptoethanol).

## Biochemical reconstitution of TBC-DEG-αβ-tubulin and TBC-DEG/TBCC-αβ-tubulin

TBC-DEG−αβ-tubulin complexes were reconstituted as described previously[28]. Briefly, recombinant TBC-DEG (5 μmol) was incubated with 5 mM GTP or GTPγS and mixed with equimolar porcine αβ-tubulin and ΔN-DARPin. Complexes were purified by size-exclusion chromatography using a Superdex 200 10/300 (Cytiva) in binding buffer (50 mM HEPES, 130 mM KCl, 3 mM β-mercaptoethanol, pH 7.0). Fractions were collected and analyzed by SDS-PAGE.

TBC-DEG/TBCC-αβ-tubulin ternary assemblies were reconstituted as previously described[28], by mixing recombinant TBC-DEG containing Q73L-Arl2 (5 μmol) with equimolar porcine αβ-tubulin, TBCC, iH5, and 5 mM GTPγS. Complexes were purified by size-exclusion chromatography using Superdex 200 10/300 (Cytiva) in binding buffer as above and analyzed by SDS-PAGE.

## Cryo-EM sample preparation and imaging

Size Exclusion chromatography purified TBC-DEG−αβ-tubulin or TBC-DEG/TBCC−αβ-tubulin complexes were obtained at 0.1 mg/ml or diluted to this concentration prior to grid preparation. Complexes were crosslinked by incubation with 20 nM BS3 for 1–2 h at room temperature, followed by quenching with 100 nM Tris-HCl (pH 7.0). Copper Quantifoil R1.2/1.3 grids (Thermo Fisher) coated with a 2-nm continuous carbon support film were glow-discharged immediately before sample application. A total of 4 μl of complex was applied to each grid, incubated in a Mark III Vitrobot (Thermo Fisher) at 20 °C and 100% humidity for 30 s, blotted at blot force 8 for 3–5 s, and plunge-frozen in liquid ethane. TBC-DEG−αβ-tubulin grids were screened, and two 5–6k-movie datasets were acquired on a Thermo Fisher Titan Krios equipped with a Gatan K2 direct electron detector. Movies were collected as 80 frames over 2 s at a total dose of 60 e−/Å². TBC-DEG/TBCC-αβ-tubulin grids were imaged on a Thermo Fisher Glacios equipped with a Gatan K3 direct electron detector. Movies were collected as 80 frames over 2 s at a total dose of 50 e−/Å².

## Single particle analysis pipeline

For each dataset, the movies were motion corrected through RELION[40] suite Motioncor2 using a 5 × 7 patch and B-factor of 150. Images were picked by LoG Picker and were subjected to 2D classification using either RELION 3.2–4.1[29,40] or Cryosparc 4.1[30]. Multiple rounds of 2D-Classification were used to remove junk particles.

For TBC-DEG-αβ-tubulin a de novo starting model was generated by the best 2D projections in RELION and was followed by three rounds of 3D classification to sort particles and remove junk or broken particles (Supplementary Fig. 2). The particles in the best classes were then combined for a 3D auto-refinement in RELION, z-flipped, and then subjected to CTF refinement and Bayesian polishing. A subsequent 3D auto-refinement led to a 3.6 Å structure containing a noisy region, which was determined to be continuous heterogeneity. These particle images were then transferred to Cryosparc, and 3D Variability Analysis[31] was performed on the particle pool. Three modes of variability were selected, intermediate mode was used containing 10 frames, and resolution was filtered to 6 Å. 3DVA separated frames of particle images in two pools with different conformations of the TBCE LRR-CapGly arm region (Supplementary Fig. 3). Frames were pooled together based on their position, subdomain organization, and general interaction interface with α-tubulin. Two pools of particles were identified with two distinct conformations. A 3D-auto refinement in

Cryosparc was performed on each pool using the most representative frame of the pool for each refinement. DeepEMhancer[41] was used to sharpen the resulting reconstructions using either the default sharpening model or the wide target model. The final processing parameters are described in Table 1. The TBCE LRR-CapGly arm conformations (Supplementary Fig. 3C−E) were stitched onto the core consensus refined TBC-DEG-αβ-tubulin map, without TBCE (Supplementary Fig. 3A, B).

For TBC-DEG/TBCC-αβ-tubulin, three de novo starting models were generated in Cryosparc through Ab-initio reconstruction by particle stacks with clear secondary structure displayed in the 2D-class averages. All starting models were subjected to heterogeneous refinement in Cryosparc[30]. A single class containing the best pool of particles was subjected to a homogeneous refinement in Cryosparc without a mask. These data were then converted into RELION format, where the following steps were performed, 3D-auto refinement, Bayesian polishing, and CTF refinement leading to a 3.6 Å structure. Continuous heterogeneity was displayed in a periphery region of the structure as well as part of the core. These regions were masked, and after conversion to Cryosparc file format, were subjected to 3D Variability Analysis[31]. Two separate 3DVA steps were performed using a different mask for each step. In the first 3DVA, a wide mask was used, including the periphery of the structure displaying continuous heterogeneity. Three components were determined, intermediate mode was used consisting of 10 frames, and resolution was filtered to 6 Å. Component #1 focused on refining the TBCC-bound particles, revealing two groups of particles in frames 0–3 and 7–10, which were subjected to local refinements in Cryosparc (Supplementary Figs. 9 and 10A, **middle right**). Component #2 focused on refining the TBC-DEG-αβ-tubulin core conformation revealing two groups of particles in frames 0–3 and 7–10, which were subjected to local refinements in Cryosparc leading to two unique states (Supplementary Fig. 9, **bottom right;** Supplementary Fig. 10B). Component #3 focused on refining the relationship between TBCC binding and TBCE LRR-CapGly arm conformation revealing two groups of particles in frames 0–3 and 7–10, which were subjected to local refinements in Cryosparc leading to two unique states (Supplementary Fig. 9, **middle left;** Supplementary Fig. 10C). Each of the three components was analyzed and comparisons were made between them to understand subunit relationships with each other. A second 3DVA step was performed with a tight mask around the core based on the outcome of one of the components of the previous 3DVA step. This 3DVA step focused on understanding the heterogeneity of the periphery region of the structure. Three components were determined, intermediate mode was used consisting of 10 frames, and resolution was filtered to 5.5 Å (Supplementary Figs. 9 and 10). Two pools of particles were identified and separated as unique TBCE states. The frames were pooled based on similar positioning and subjected to local refinements in Cryosparc to achieve final reconstructions. All structures were sharpened using DeepEMhancer with a wide target model. The final processing parameters are described in Table 1. Fourier Shell correlation (FSC) and resolution (Res) maps for the refined 3DVA component maps are presented in Supplementary Fig. 11. The scheme for assembly of the two composite maps for TBC-DEG/TBCC-αβ-tubulin state 1 and state 2 is presented in Supplementary Fig. 12A. The scheme represents relationships of change in different regions in relation to TBCC domains binding. Two final composite maps are assembled and presented (Supplementary Fig. 12B).

## Model building and refinement

The two TBC-DEG-αβ-tubulin binary composite Cryo-EM maps were built using a combination of ISOLDE, Coot[42], and PHENIX[43] starting with the AlphaFold3 models for budding yeast TBCD, TBCE, and Arl2 and the porcine αβ-tubulin (PDB ID:1FFX). AlphaFold3 models were initially fit in ChimeraX and morphed using ISOLDE. The two binary state maps for TBCE (Class1 and Class2) map densities were built based

on the placement and minor changes to the LRR and CapGly Alpha-Fold3 models, but manual placement of the 3HB helices and connection between LRR and CapGly was necessary. For the TBC-DEG binary structures, the sequence registry was first identified by placing bulky side chains. However, due to weaker density in these structures, all side chains were truncated to Cβ. The side chains of residues within the truncated TBCE LRR and CapGly regions were replaced with alanine because of weak or discontinuous density. The models were subjected to cycles of Coot-based manual building of loops and side chain corrections and real space refinements in PHENIX. The final model validation was performed in PHENIX (Table 1). Models and maps are deposited into the RSCB and EMDB as described in the Data availability statement and Table 1. Figures were generated using ChimeraX[44].

Two composite TBC-DEG/TBCC-αβ-tubulin ternary maps were generated by stitching mask-based regions of maps generated from 3DVA components as described in Supplementary Fig. 8. AlphaFold3 models for the two states were used as starting points for building, and a similar pathway as described for model building the TBC-DEG-αβ-tubulin binary structures was used. The two ternary state maps for TBCE (Class1 and Class2) map densities were built based on the placement and minor changes to the LRR and CapGly TBCE models from the AlphaFold3 models. Manual placement of the 3HB helices and connection between LRR and CapGly was necessary. Similarly, the TBCE LRR and CapGly domains were modeled as poly-alanine while retaining the backbone trace to preserve the sequence registry inferred from the well-defined C-terminal domain of TBCE. The C-terminal region of TBCC exhibited reduced density, and side chains in this region were truncated to Cβ. The C-terminal region of TBCE was well resolved and modeled with full side chains in the ternary structures. The models were subjected to cycles of PHENIX real-space refinement and Coot-based manual building of loops and side chain corrections (Table 1). Models and maps are deposited into the RSCB and EMDB as described in the Data availability statement and Table 1. Figures were generated using ChimeraX[44].

## AlphaFold3 model predictions

To determine TBC-DEG-αβ-tubulin AlphaFold3 models, sequences for the budding yeast TBCD, TBCE, Arl2, porcine α-tubulin, porcine β-tubulin, two GTP and one GDP molecule were entered into a single multi-subunit determination using the AlphaFold3 server (www.alphafoldserver.com)[45]. A single representative model is presented in Supplementary Fig. 7A with the moderate to high confidence pIDDT values per residue displayed (Supplementary Fig. 7A) and their corresponding PAE matrix (Fig. 7B) with accuracy of residue position error. To determine TBC-DEG/TBCC-αβ-tubulin AlphaFold3 models, sequences for the yeast TBCD, TBCE, Arl2, porcine α-tubulin, porcine β-tubulin, TBCC, three GTP molecules were entered into a single multi-subunit determination using the AlphaFold3 server (www.alphafoldserver.com). All five AlphaFold3 models were comparable in PAE value. A single representative model is presented in Supplementary Fig. 16A with the moderate to high confidence pIDDT values per residue displayed (Supplementary Fig. 16A) and their corresponding PAE matrix (Supplementary Fig. 16B) with accuracy of PAE per residue.

## Reporting summary

Further information on research design is available in the Nature Portfolio Reporting Summary linked to this article.

## Data availability

Cryo-EM maps and models are available in the Electron Microscopy Database (EMDB) with the EMBD-IDs: EMD-47949, EMD-47954, EMD-47947, EMD-47948. The corresponding atomic coordinates for models are available at the Protein Data Bank (PDB) with the accession numbers, PDB-ID: 9EDT, 9EEB, 9EDR, 9EDS, respectively. The work also utilized the following coordinates for model building and comparisons: 1FFX, 4DRX, and 6GWD. Source data are provided with this paper.

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

## Acknowledgements

We thank the Bay Area Cryo-EM consortium led by Prof. Eva Nogales (Molecular Cell Biology, UC-Berkeley) and the UC-Davis BioEM facility for Cryo-EM data collection support. Large Scale Cryo-EM data were collected at UC-Davis and the Cryo-EM facility at UC-San Francisco with support from Dr Alexander Mysanikov (Biochemistry and Biophysics, UC-San Francisco). We thank Dr. Camille Scott and the UC-Davis High-Performance Computing Facility (HPCCF) for computational HPC infrastructure building and support. We thank Dr Stanley Nithianantham (Molecular Cellular Biology, UC-Davis) for the preliminary biochemical studies. We thank Prof Jeffrey K Moore (Cell Developmental Biology, University of Colorado, Anschutz Medical Campus) for advice and suggestions and for the critical reading of this manuscript. We thank Prof Richard McKenney, Prof Jonathan Scholey (Molecular Cellular Biology, UC-Davis), and Prof Ahmet Yildiz (Molecular Cell Biology, UC-Berkeley) for comments on the manuscript. J.A.B. acknowledges funding support from the National Institutes of Health (GM110283 and GM158334).

## Author contributions

A.T. purified and assembled TBC-DEG/TBCC-αβ-tubulin complexes, collected cryo-EM data, determined and refined all single particle cryo-EM structures, built, refined, and validated all models for assemblies, prepared figures, wrote and edited the manuscript. Z.W. purified and assembled TBC-DEG-αβ-tubulin complexes, collected cryo-EM data, and determined initial single particle cryo-EM structures. B.S. built and refined initial models. F.G. supported cryo-EM grid preparation, cryo-EM screening, and large-scale cryo-EM data collection. J.A.B. planned and managed the project, trained scientists, obtained funding for the project, prepared assemblies, prepared figures, and wrote and edited the manuscript.

## Competing interests

The authors declare no competing interests.
