## [Transparent Peer Review file · Nature Communications]

Cryo-EM structures of the tubulin cofactors reveal the molecular basis of alpha/beta-tubulin biogenesis

Corresponding Author: Dr Jawdat Al-Bassam

Version 0:

Reviewer comments:

Reviewer #1

(Remarks to the Author)

The work by Taheri et al describes the first cryo-EM structures of tubulin cofactors TBCD, TBCE and Arl2 (TBC-DEG) bound to an alpha/beta tubulin heterodimer as well as the same complex bound to an additional tubulin cofactor TBCC. Tubulin cofactors are fascinating and play a fundamental role in eukaryote organisms. Their structures have the potential to elucidate the important question of how they work together to generate functional tubulin heterodimers.

Currently I am afraid that the cryo-EM maps that accompany the manuscript do not appear to justify the conclusions that are drawn. The overall map resolution is not consistent with the claimed 3.6Å. This makes observations such as which nucleotides present or the presence of tubulin rotations unreliable. Furthermore the half maps show weak density for many parts of the structure and even the DeepEMhancer maps show poor fits with the models. The current maps therefore do not fully support the proposed structural rearrangements during the cycle.

I would encourage the authors to either adjust their manuscript to describe what they can see with the current maps or to work further on the structure.

Major Comments:

1) GTP density in Arl2 : Density not clear to distinguish between GDP or GTP in the TBC-tubulin structures. Only structure with defined GTP density is the TBC_tubulin_TBCC_state1.pdb. The nucleotide conformation is also different among the structures, likely due to limited resolution.

2) Figure2: Panel G,J,K- most side chains not visible in map. Misleading.

3) Figure2: Panel K - Residue labels for the blue chain (Ubq) are incorrect when compared to pdb (I looked at TBC_tubulin_state1.pdb)

4) The TBCE LRR-CapGly density is poor, probably in the 7-8 Å regime at best. Based on the current map of TBC_tubulin_TBCC_state2 it is very difficult to judge whether the positioning of this segment is accurate due to the fragmented density. An unsharpened map or a map filtered down to 7-8Å might be helpful for this purpose.

a) There is not much density for TBCE in the half maps or unsharpened map. The only map that shows some density in this region is the deepEMhancer output map. Even in that map the correlation between the model and the map seem very low.

b) The TBCE region is better defined in the TBCC containing complex but even then the map quality is worse than the reported local resolution of 5-7 Å in these regions.

5) I don't see the tails binding the TBC-DEG in the structure as per this statement 'The C-terminal tails of both α - and β -tubulins are bound to TBC-DEG and fully occluded in the TBC-DEG- $\alpha\beta$ -tubulin structures'. The tail of alpha-tubulin is built

but the density is poor and it still does not interact with anything except itself.

6) The TBCC modelled bound to Arl2 is different in the two states presented in this paper. The density for TBCC is poor in both maps which likely is the reason for this. The figures showing the local resolution in this region suggests its at 3.5 angstrom. The density does not suggest that.

a) the authors need to explain in detail how their model building was done for all such regions.

7) There is rotation of the alpha vs the beta tubulin in the TBCC state 1 vs the model without TBCC but the interfaces between the alpha and beta subunit remain the same. The rotation if any comes from small deviations in the non-interacting regions. This is in contrast to the conclusions in the manuscript.

8) In Figure S3, the FSC curve labels appear to be incorrect. Based on the appearance of the graph, the FSC curves in panel E and F seem to be generated in RELION however the labels for each line seem to follow the convention of Cryosparc which is different from RELION and therefore incorrect.

Reviewer #2

(Remarks to the Author)

Version 1:

Reviewer comments:

Reviewer #3

(Remarks to the Author)

Overall, this study provides intriguing structural insights into how TBCC, TBCD and TBCE interact with alpha-beta-tubulin, but several clarifications are still needed for certain interpretations and mechanistic conclusions. Addressing these points will strengthen the manuscript's clarity and claims.

Major Comments:

1) Line 145: The text states: "In this conformation, TBCE is fully bound to alpha-tubulin and alpha-beta intradimer interface." However, the figures and models do not show anything bound to the interdimer interface. Please clarify or provide evidence supporting this statement, or revise the text accordingly.

2) Lines 204–211 (Figure 2D, Figure S4F–G): The text describes two states for TBCE's engagement with alpha-tubulin:

a. State 2: The LRR domain engages the alpha-tubulin lateral polymerizing interface.

b. State 1: The CapGly domain engages the alpha-tubulin C-terminus, while the LRR retracts.

However, the figures do not clearly show CapGly involvement. Instead, it appears that the LRR remains bound to alpha-tubulin in both states. Please clarify or update the figures to reflect the proposed binding modes.

3) Line 320 (Figure S10G): The text mentions that TBCC-C binding to Arl2 GTPase induces changes in Arl2 (switch I and II) and causes TBCD to create a tighter β -tubulin. However, these structural changes are not clearly illustrated in any main or supplementary figure, and the panel G cited in Figure S10 is missing. Please modify the figure(s) or the text to show these changes more explicitly.

4) Line 336 : The authors state that: "The TBCE LRR-CapGly rotation shifts β -tubulin likely by pulling its C-terminal tail (Figure 3D-E, Figure 5B). However, due to the low resolution of these regions of the cryo-EM structures we were unable to observe the β -tubulin C-terminal tail. " These seem to be two contradictory statements, if you cannot see the C-terminal tail bind TBCE then how do you propose that TBCE pulls it. The authors should remove this or provide better explanation for their hypothesis/model.

5) Line 363: The manuscript states that Arl2 GTP hydrolysis is the source of energy and that TBCC activates TBC-DEG as a platform for alpha-beta-tubulin assembly and degradation. It is not clear of what structural evidence supports this. Clarifying which results indicate this would be useful or revise the statement to reflect uncertainty. Also clarify/explain if this interpretation is also based on previous observations.

6) Line 365: The text describes TBC-DEG overcoming alpha-beta-tubulin stability by deforming the intradimer interface at the N-site GTP, ultimately destabilizing alpha-beta-tubulin. Meanwhile, TBCC is described as catalyzing TBC-DEG transitions that reform the alpha-beta-tubulin intradimer interface.

Consider comparing the alpha-beta-tubulin conformation in TBC-DEG vs. TBCC-bound structures rather than relying solely on AlphaFold-based comparisons. Based on the figures presented, the alpha-beta-tubulin conformations don't seem to be considerably different between the two structures (TBC-DEG vs. TBCC-bound) which confuses me as to why then TBC-

DEG is proposed to be deforming whereas TBCC bound state to be the opposite. If the observed differences between the tubulin heterodimers in these states are minor, you might need to change your discussion or explain why you think those minor differences are important and not just a result of the flexibility of the complexes.

Further I think comparing AlphaFold predictions to experimental models for deformations is not the best comparison as we do not know if those fine details in the predictions are real or an artefact. So, making mechanistic conclusions from them is risky!

7) Line 373: The statement "The AlphaFold 3 predicted models reveal TBCC binding reverses the deformation of the alpha-beta-tubulin dimer observed in TBC-DEG-alpha-beta-tubulin states" should be softened to indicate that these results are consistent with the experimental data. Avoid implying that AlphaFold predictions alone confirm a mechanism. See above comment as well.

8) Many of the low-resolution regions associated with TBCE exhibit poor validation scores (such as Qscores). Consider truncating side chains in these regions to improve the overall model quality, and consider making the corresponding AlphaFold models publicly available for anyone interested in examining the predicted side chains in more detail.

Minor comments:

- 1) Figure S3: The FSC labels still appear to be incorrect. Lines in panels B, E, and G look like they were generated in RELION, yet the labeling follows CryoSPARC conventions. For example, the red line is not "No-Mask" but rather a phase-randomized map. Please confirm each line's meaning and correct the figure legends accordingly.
- 2) Figure S4: Typo in panel "F": "Ubq" is labeled as "ibq."
- 3) Line 192: Typo in parentheses: it should be (I, II, and III).
- 4) Figure S6: Many of the side chains for TBCD and E are not clearly visible but are shown in the figure. If side-chain information comes from AlphaFold models then the figure legend should state this.
- 5) Line 243: There is a premature full stop '.' Please fix the punctuation.
- 6) Methods: In several instances, the pH of the buffers used is not mentioned. Please add these details wherever applicable.

Reviewer #4

(Remarks to the Author)

I have been asked to arbitrate Taheri et al.'s point-by-point reply to Reviewer #2's original comments.

Regarding point 1): The authors have addressed the Reviewer's concern about map renderings.

Regarding point 2): The clash scores remain unusually high. After examining the maps and models to-be-deposited, there could be several explanations: a) the models were refined against DeepEMhancer processed maps, which the consensus in the field is that this should probably be avoided; and/or b) the resolutions achieved lack accurate density for most side chains, no matter the post-processing approach. The authors must clarify which version of the final map(s) was used for model refinement; if clash scores remain >10 after refining against standard RELION/phenix auto-sharpened maps (recommended), side chains are likely not supported by density and should be truncated to poly-A for model deposition and any description of experimentally-observed side chain interactions should be removed from the paper.

Regarding point 3): The authors have addressed the Reviewer's concern about missing map and model statistics.

Regarding point 4): Although I understand it would require additional experiments, the authors have not entirely addressed the Reviewer's reasonable concern that DARPin / iH5 might occlude or interfere with structural changes in tubulin or other parts of the TBC. Indeed, the authors claim themselves that iH5 was needed to resolve TBCC in the first place, indicating that these proteins do have the potential to alter the TBC's conformational state(s). Without additional data to control for the use of these capping factors, the interpretation of TBC conformational changes is limited and should be described in a more reserved manner.

Regarding point 5): The authors have addressed the Reviewer's concern about naming conventions for the various TBC conformations.

Regarding point 6): The swivel of the CAP-Gly domain is not at all clear from Figure 1E. Is the CAP-Gly domain density even still resolved in state 2?

Regarding point 7): While the authors do include the details requested by Reviewer #2, the figures remain incredibly busy and difficult to follow clearly. I have the feeling that many figure panels are redundant and unnecessarily labeled. A few of the many examples of this are: re-labeling each helix number in e.g. Figure 2C, far right; also the coloring of negative/positive residues on top of colored interacting surfaces in the same figure panel, etc.

Another point is that the legend for video 1 claims that the cryo-EM map is shown in transparent surface. I am pretty sure this is a surface representation of the model. This is quite misleading and the authors' general use of models as transparent density-like surfaces in the figures leads to additional distractions in their figures, since the reader has to additionally understand the difference between experimental density and the underlying model, which has limitations based on local

resolutions achieved. In my opinion, such representations should be entirely avoided.

Regarding point 8): The authors have explained how their composite maps were generated. However, given the fact that only through 3DVA were some parts of the complexes resolvable by cryo-EM, I would strongly advise uploading all initial maps used to generate composite maps to the EMDB as "maps associated with the main deposition". Related to this, if not already included in the deposition, I would strongly suggest the authors deposit the half maps and masks used in final RELION refinements, so that non-DeepEMhancer sharpened data can be made available with the study.

Regarding point 9): The authors have addressed the Reviewer's concern about nucleotide density rendering. However, related to point 2) above, the authors could be more careful in their assignment of e.g. GDP in the exchangeable site of β -tubulin, which, although likely, is not always clear from the density maps provided.

Regarding minor point 1): The authors have addressed the Reviewer's concern.

Regarding minor point 2): The figures remain quite "busy" with still a lot of redundant information (see point 7 above), in my opinion.

Regarding minor point 3): While all steps are clearly labeled, the final maps used for model building or composite map generation are not clear from the figure.

Regarding minor point 4): The Reviewer #2 was correct in suggesting using a range of 3-6 Å for local resolution estimates. The current range makes it seem like most of the density is at ~3 Å, which does not appear to be the case after inspecting the individual maps.

Regarding minor point 5): The authors have addressed the Reviewer's concern.

Regarding minor point 6): While all steps are clearly labeled, the final maps used for model building or composite map generation are not clear from the figure.

Reviewer #5

(Remarks to the Author)

The original Reviewer #2 was unable to continue reviewing this manuscript and I was invited to act as an arbitrating referee to judge the original referee concerns, as well as the author responses.

Overall, the comments of Reviewer #2 were valid and addressed a couple of major and important technical issues in the original manuscript version. The revised manuscript version addresses these issues partially, but not comprehensively.

Please find my comments related to the individual points below ('new comments').

Reviewer #2

The manuscript by A. Taheri et al. presents cryo-EM structures of tubulin cofactors bound to $\alpha\beta$ -tubulin, specifically the TBC-DEG- $\alpha\beta$ -tubulin and TBC-DEG/TBCC- $\alpha\beta$ -tubulin complexed, which have not been reported before. By comparing conformational variations, the authors report a transition state in $\alpha\beta$ -tubulin biogenesis, and they propose the molecular basis on how TBC-DEG/TBCC cofactors regulate this process. Their structures are new to the field, and the overall research topic is interesting. However, there are essential technical issues that the authors need to address:

1) The reported map resolutions are in the 3.7-3.8 Å range; however, the provided map renderings do not reflect the expected level of structural details for this resolution. The authors are suggested to render their structures at higher threshold to better illustrate the structure features. This would improve the visibility of subunit boundaries, allowing for clearer identification of interaction interfaces. Otherwise, it is challenging to accurately visualize and interpret the subunit interactions and measure the domain movements.

Author response: We thank the reviewer for their suggestions regarding the presentation of the atomic model to electron density fits. We have revised the presentation of Figures 1-4 and Figures S4 and S12 to demonstrate the resolution and quality of maps and built models into the structures as suggested. We believe the revised figures accurately reflect the resolution of the cryo-EM maps presented. We further present the revised atomic model validation statistics that have since improved (discussed below). We also present the final the PDB reports for the structures after submission to the RSCB (see attached reports).

New comment: This is a very important comment. Rendering the cryo-EM densities at appropriate threshold is central to assessing the quality of the reconstructions and the fit of the atomic model. While the densities have been rendered at slightly higher threshold in the revised manuscript version, it is still very challenging to recognize structural features consistent with the reported resolution of the density. It would be highly desirable to update the figures once more at substantially higher threshold level, which allows visualization of individual side chains.

2) The quality of the models also needs significant improvement. As indicated in Table 1, the Clash Scores ranges from 19.03

to 26.02, which is unacceptably high for well-refined models. In multiple instances, the model does not appear to fit properly within the cryo-EM density, further suggesting that additional refinement of the models are necessary.

Author Response: We agree with the reviewer's concerns. In the revised manuscript, we have rebuilt atomic models and refined these models which now lead to improved clash scores for each of the structures presented in the manuscript. The PDB reports are provided for the reviewers to address the quality of these maps. The predicted AlphaFold 3 models for TBC-DEG:ab-tubulin and TBC-DEG/TBCC-ab-tubulin (see above discussion) support most of the major conclusions about the assembly of the tubulin cofactors and their interactions with a and b-tubulin, as well as the TBC-DEG and TBCC impact on the ab-tubulin dimer configuration.

New comment: This is a very valid comment. The clash score of the refined models has marginally improved, but it is still comparably high for the resolution claimed and further improvement would be desirable.

3) In Table 1, please provide the Map sharpening B factor value. Additionally, the unit for Electron Exposure should be ($e^-/\text{Å}^2$), instead of the current " $(e/\text{Å}^2)$ ". Moreover, the authors are suggested to provide the map and model evaluation profiles from the PDB/EMDB validation reports for a more comprehensive assessment of the map and model quality.

Author Response: In the revised manuscript, we present the rebuilt atomic models and the model to map comparisons (Figures S4 and S12). The revised validation statistics and details should address the concerns of the reviewer. We have now provided the PDB validation reports as well.

New comment: While the authors have provided the PDB validation reports, Table 1 has not been updated as suggested by the original Reviewer #2. The map sharpening B factor is an important parameter that should be reported for each cryo-EM density.

4) In the two complexes, different binding factors were included to address the preferred orientation issue: Δ N-DARPin in the TBC-DEG- $\alpha\beta$ -tubulin complex, and α -rep iH5 in the TBC-DEG/TBCC- $\alpha\beta$ -tubulin complex. By comparing the conformations of $\alpha\beta$ -tubulin in these two complexes, the authors conclude that there is a 4-degree twisting of α -tubulin, while β -tubulin is held in place by TBCE. However, it is unclear how the authors ruled out the possibility that the observed α -tubulin twisting is not due to the different binding factors, particularly in the TBC-DEG/TBCC- $\alpha\beta$ -tubulin complex, where iH5 interacts directly with α -tubulin. To clarify this, the authors should use the same binding factor in both complexes to address the preferred orientation issue and compare the conformational changes.

Author response: We appreciate this question, and we present a revised supplementary figure that addresses the comparison indicated by the reviewer (Figure S13C). The structures of ab-tubulin bound to DARPin and iH5 has been determined by x-ray crystallography and published previously. These structures show an identical conformation in comparison to native ab-tubulin dimer, in terms of the positioning of a tubulin with respect to b-tubulin. We present a comparison of these structures to our TBC-DEG:ab-tubulin and TBC-DEG/TBCC:ab-tubulin models. These comparisons show that iH5 or DARPin binding does not impact the conformation of a and b-tubulin, but rather TBC-DEG binding alters a-tubulin. We were unable to use the same tubulin capping proteins in both structural studies, since we found that DARPin affected the binding of the TBCC complex, likely due to shifts in the TBCE C-terminus and TBCE-ubq domains that are induced by TBCC binding, leading to the occlusion of the DARPin binding site. The tubulin capping proteins were required in these structural studies as the protein complexes aggregated on the cryo-EM grids without them.

New comment: The comparison of ab-tubulin structures bound to DARPin and iH5 as determined by X-ray crystallography strongly supports the authors' conclusion that TBC-DEG binding rather than DARPin or iH5 binding alters the ab-tubulin arrangement. However, since the conformational space of complexes might be restricted after crystallization, the original Reviewer #2's concern cannot be entirely ruled out with this comparison. Please consider addressing this possibility in the discussion section of the manuscript.

5) To avoid confusion, the authors are advised to consistently refer to the four maps as "states 1-4", rather than using both "classes" and "states".

Author response: We thank the reviewer for these suggestions. And we have revised the naming of the states in throughout the manuscript to address this. We refer to two states of TBC-DEG:ab-tubulin, State 1 and State 2 and two states of TBC-DEG/TBCC:ab-tubulin termed State 1 and State 2.

New comment: This comment seems to have been sufficiently addressed.

6) The authors state, "TBCE exhibits two distinct conformations of its CapGly and LRR domains (Figure 1C; Figure S4F-G). In the first conformation (class 1), the TBCE 3HB and LRR are oriented vertically and the CapGly resides horizontally and alongside the LRR (Figure 1C; Figure S4F-G). In this conformation, TBCE is retracted from binding a-tubulin". However, based on Video 1 and the map shown in Fig. 1A, there appears to be clear contact between TBCE and α -tubulin in both states. Please clarify this discrepancy.

Author response: We agree with the reviewer's point regarding the confusion and have revised this section to clarify this point. The TBCE-LRR contacts α -tubulin, but this contact is near the base of α -tubulin but in the second state this contact is fully extended along the α -tubulin surface. The difference is mediated by a conformational change in the N-terminal region and a swivel of the CapGly domain. WE now present this in Figure 1E of the revised manuscript.

New comment: This comment seems to have been sufficiently addressed.

7) On P.4, the authors describe: "Conserved residues in the intra-HEAT turns of TBCD H1, H2, H3, H5, H6, H8, and H10 bind residues at the lower surface of the Arl2 GTPase through ionic and hydrophobic interactions (Figure 2A, E-F; Figure S14D-E; Figure S14DE, G, I-J). Conserved TBCD Lys and Arg residues bind conserved Arl2 Asp, Gln, and Glu residues (Figure S11, S14 D-E, G; Figure S14D-E). Hydrophobic packing of conserved Leu, Phe, and Trp residues in TBCD and Arl2 are interspersed amongst the ionic interactions (Figure 2E-F; Figure S14D-E, G; Figure S16I-J)." However, the related Fig. 2E-K are too busy and difficult to interpret. As suggested, please further refine the models to high quality, then thoroughly analysis the the interaction network between neighboring subunits. Sepcificlly, consider analysing the H-bonds/Salt bridges, as well as electrostatic and hydrophobic/hydrophobic surface properties to better illustrate these interactions.

Author response: We thank the reviewer for their suggestions and have revised both the models and the presentation of the interaction interfaces to address their concerns. In addition, we have included AlphaFold 3 models for the TBC-DEG-ab-tubulin and TBC-DEG-ab-tubulin- TBCC complexes (Figure S7 and S14) which reveal matching interaction sites to the experimentally derived models from cryo-EM data. Figures 2 and 4 have been revised and now focus on the general interfaces while the interaction interfaces are now presented in supplementary figures S5-6 and S15.

New comment: While the presentation in Supplementary Figs improved, the interfaces in Figures 2 and 4 are not yet very clearly presented. In particular, it is very challenging to differentiate subunit-based surface coloring from electrostatic / hydrophobic surface coloring. Please consider visualization of the surface properties on otherwise uncolored surfaces in Figures 2 and 4.

8) The authors state, "Using these conformations and the subunit binding relationships to TBCC, we generated two composite maps for two distinct conformations of the ternary assemblies (Figure S8)." However, there are multiple conformations present in different regions of the complex, and the authors performed separate 3DVA/refinement on individual local regions. It is unclear what criteria were used to integrate these separate maps, which reflect such diverse conformational and spatial information, into two consensus maps. Could the authors clarify the justification behind merging these diversified data into only two representing maps?

Author response: We understand the reviewer's concern. We have revised the results to describe how the two composite maps were generated. The composite maps were generated based on the impact of binding of TBCC-C and TBCC-N on transitions on the TBCD and TBCE conformations. The first composite includes TBCC-C binding to the TBC-DEG-ab-tubulin induces a transition the TBCE LLR-CapGly arm. In the second composite both TBCC-C and TBCC-N bound suggesting unique conformational transitions associated with its binding on the TBCD and TBCE LRR-CapGly arm. The two composites reveal the clearest step wise transitions induced by the two-fold binding of TBCC onto different regions of TBC-DEG.

New comment: This is a very important comment. While the revised results section now provides more information as to how the two global states were assembled, an additional figure clearly illustrating the correlation between structural differences represented by the four components would be very helpful for clarification.

9) Fig. S9I, regarding the identification of GTP or GDP, the current rendering is cluttered with a busy background, and the segmented density appears too large. As a result, it is challenging to distinguish whether the bound nucleotide is GTP or GDP.

Author response: We have revised the presentation of nucleotide densities to be included in Figures S4 and S12. We can clearly observe the nucleotide densities in the maps and assign di- or tri-phosphate states in each of the three nucleotide pockets, Arl2, E-site and N-site. We believe the assignment is accurate and matches the protein conformations expected based on the AlphaFold 3 models and previous structural studies of tubulin dimers.

New comment: Even in the revised version of the manuscript, the segmented density is rendered at too low threshold level, making it challenging to distinguish GDP from GTP. Please show at substantially higher threshold level. Please explain in the legends/methods section how the density segment attributed to nucleotides was derived/segmented.

Minor point:

1) Fig. S2-3, please also show the FSC and local Resolution etc. for the core map.

Author response: We have included the FSC for the core map in the revised manuscript Figure S3.

New comment: This comment seems to have been sufficiently addressed.

2) Fig. 1A-B contain a lot of redundant information, but do not present the class 2 structural features. I recommend including the maps and models of core, class1, and class2, and move the key structural analysis from supplementary figures into the main figure.

Author response: We thank the reviewer for this suggestion. We have revised the presentation to include the revised versions of these figures.

New comment: This comment seems to have been sufficiently addressed.

3) Fig. S2 does not clearly illustrate the processing procedure, making it extremely difficult to understand how the two datasets were processed. Please re-render the reconstruction procedures to clarify this.

Author response: We have relabeled the processing procedure more carefully to explain the steps presented.

New comment: Even in the revised version of Fig. S2, it is very challenging to follow the image processing procedure. Please consider using color-coding for individual processing branches to deconvolute the figure.

4) In Fig. S3C-D and Fig. S8A-C, adjust the local resolution rendering scale to start from 3 Å instead of 2 Å. A range of 3-6 Å should be sufficient.

Author response: Considering that 3.6 Angstrom is the average resolution of the core we believe the current resolution range is sufficient to present the Res map colors for the maps presented.

New comment: I agree with the original Reviewer #2 that color coding from 3-6 Å may represent the resolution range of the densities more accurately. Please consider following the original Reviewer #2's suggestion.

5) In Fig. S4A-E and Fig. S9, the high-resolution structural features are not visible. Please re-render them with a higher density threshold

Author response: We thank the reviewer for this suggestion. These figures have been replaced.

New comment: This comment seems to have been sufficiently addressed.

6) Fig. S6 shows numerous reconstruction tracks, but it is unclear which portions were used for analysis and which track led to the final map. Please clarify how the composite map was assembled, specifying which sections came from where and the rationale behind their inclusion. Please re-render the reconstruction procedures to clarify these points

Author response: We have altered the presentation of the arrows to improve the clarity of the directions of processing flow and clarify the procedures.

New comment: While the image processing procedure in Fig. S9 is clearly represented, the original Reviewer #2's concern remains. It is not clear which segments of the different reconstruction tracks were used to generate the composite densities. Please consider highlighting those segments using masks.

Version 2:

Reviewer comments:

Reviewer #3

(Remarks to the Author)

The authors have improved their description of their structures. These structures will be very useful for the many researchers. However, some of my comments related to interpretation of the structures were not addressed appropriately (see comments below). Further, in general I would suggest the authors to not speculate in the results and only state observations clearly supported by their experimental data and can instead explain their model in the discussion. This will help distinguish what their data shows versus what they think their data suggests.

Comments not addressed properly:

1) My old comment: Line 320 (Figure S10G): The text mentions that TBCC-C binding to Arl2 GTPase induces changes in Arl2 (switch I and II) and causes TBCC to create a tighter β - tubulin. However, these structural changes are not clearly illustrated in any main or supplementary figure, and the panel G cited in Figure S10 is missing. Please modify the figure(s) or the text to show these changes more explicitly.

Authors response: Figure S13A–B illustrates the changes in the TBCD interface with β -tubulin in the various structures. In panel B, from left to right, the changes in TBCD's tightened grip on β -tubulin are observed. We have clarified and removed the discussion of changes in Arl2 upon TBCC binding, as the differences are not crucial for the presentation.

My new comment: The current figure is not helpful to judge that TBCD has a tightened grip. Comparing the structures myself, it actually doesn't seem that the binding interfaces between TBCD and beta-tubulin have changed significantly. So I am actually not sure on what basis do the author say this and include it in their final model.

2) My old comment : Line 336 : The authors state that: "The TBCE LRR-CapGly rotation shifts α -tubulin likely by pulling its C-terminal tail (Figure 3D-E, Figure 5B). However, due to the low resolution of these regions of the cryo-EM structures we were unable to observe the α -tubulin C-terminal tail. " These seem to be two contradictory statements, if you cannot see the C-terminal tail bind TBCE then how do you propose that TBCE pulls it. The authors should remove this or provide better explanation for their hypothesis/model.

Authors response: The low resolution of the electron density prevented experimental modeling of the α -tubulin C-termini. However, the location of the TBCE CapGly in proximity to the α -tubulin C-terminus suggests that it likely physically binds the α -tubulin C-terminal tail. We have revised the text to more cautiously describe the interaction and its role in altering α -tubulin.

My new comment: The authors still keep stating that the C-terminal tail binds TBCE (line 211, 219, 281) but they have no experimental evidence for this. The resolution of TBCE near the C-terminus of Alpha-tubulin is low resolution to put it mildly. This point is a speculation and should solely be stated as such in the discussion.

3) General comment about last results section: There are a few statements in this results section that are not supported by the experimental structures and are an over interpretation of the data. I am stating them below and they seem to be more appropriate for the discussion as they are speculative and therefore should be stated as such in just the discussion -

Line 328: TBCC-C binding to the Arl2 GTPase engages loops around the GTP binding 329 pocket (Figure S15D), causing TBCD to create a tighter interface with β -tubulin in comparison to 330 the TBCD interface with β -tubulin in the binary assembly

Line 344: The TBCE LRR-CapGly rotation shifts β -tubulin potentially by pulling its C-terminal tail (Figure 3D-E, Figure 5B).

Line 347: Comparison of 348 State II to State III shows a 1.5 Å twist of the β -tubulin position at the intradimer interface, likely caused by TBCE LRR-CapGly arm rotation (Figure 5B-C).

Reviewer #4

(Remarks to the Author)

I will focus again only on assessing whether the authors have appropriately addressed the initial comments of Reviewer #2, as originally requested.

On the basis of the author's rebuttal, I unfortunately cannot agree that they have fully done so.

Regarding the reply to point 2), and as clearly re-iterated by point 7 from Reviewer 3, all of the commonly used metrics (not just clash scores, but also Q-scores, etc.) remain surprisingly low for a structure reported at ~ 3.7 Å resolution. The studies cited by the authors have noticeably higher average Q-scores than calculated in the PDB validation reports, confirming that something is "off" about the authors' maps and/or models. It is still unclear to me exactly which map was used for model refinement. Using DeepEMhancer-sharpened maps to real-space refine models is certainly not standard practice and it is well-appreciated across the entire cryo-EM field that this should be avoided. Further, the authors claim "All experimental model side chains were built and can be observed in the maps including Figure S6. The smaller side chains required lower contours than some of those shown in figure S6. The experimental electron density was used to place the side chains." Visual inspection of the maps and models does not support this claim for most side chains, and raises further concerns about model interpretability of side chains and even secondary structure / domain movements described in the paper. Irrespective of whether other studies have similarly-poor clash scores, it is my opinion that any part of the model - including side chains - that is not well-supported by experimental density should be removed. The study contains only this structural data and does not perform any other validations of their side chain-containing models, so particular care in model building and interpretation is certainly warranted. This remains a major concern that is also shared by several other reviewers. To be clear, I am less concerned about the fact that the maps and models need improvement - these appear to be challenging specimens - rather, I am very concerned about the mechanistic interpretations made based on these data, which are only seemingly (and questionably) supported by the cryo-EM and AF3 modeling.

Regarding the reply to point 4), I respectfully disagree. The authors cannot rule out potential roles of capping protein binding to the conformational landscape explored by α/β -tubulin during "biogenesis" without other supporting data. This remains a major concern that is also shared by several other reviewers.

Regarding the reply to minor point 4), the point has been made clear by several reviewers now that the coloring and resolution range in this supplemental figure is not appropriate. The authors' refusal to address this is concerning.

Reviewer #5

(Remarks to the Author)

Overall, the authors have done a good job in addressing my comments.

In regards to the relatively high clash scores, the authors argued that other models with comparable clash scores have been published before. This is true, but does not improve the quality of the models, which could have been desirable.

The following points still should be addressed in the final revised version of the manuscript:

- 1) If no singular map sharpening B factor can be provided, please at least provide the range of map sharpening B factors used for each cryo-EM density in Table 1.
- 2) When showing cryo-EM density of nucleotides, the authors should clearly indicate in the figure legends how the density segment was derived. Is it an 'omit map', after excluding density explained by the protein components (which would be the least biased approach)? Or is it segmented based on the model of the nucleotide itself? If so, was a di- or tri-nucleotide used?

Dear Editor and Reviewers,

We are submitting a revised version of our manuscript entitled “Cryo-EM structures of the tubulin cofactors reveal the molecular basis of alpha/beta-tubulin biogenesis” in which we address the concerns raised by the reviewers. We thank both reviewers for their detailed suggestions and have revised the manuscript to address the concerns raised in the following areas:

1) Addressed data processing and cryo-EM map resolution presentation concerns

We have provided more details about the data processing pipelines, improve their presentation clarity, and have revised the resolution presentation of different regions of the structure figures to display the variable resolution of the core regions versus the mobile TBCE arm and TBCC-C domain regions. We present a revised description and naming of the two pre-catalytic and two post-catalytic states that is more consistent throughout the revised manuscript.

2) Revised the model building and resolved model validation concerns

The reviewers raised concerns about the clash scores and quality of the atomic model building. We have revised the atomic model building and have validated and submitted all the models to the RCSB. We present the resulting PDB reports as well (attached to this submission). The revised atomic model building utilized already known information about the structures, but also incorporates information from predicted AlphaFold models (see below) which agree with the atomic models derived from the cryo-EM structures. The AlphaFold 3 models (see below) have aided in the revision of the atomic models including the orientation of the interacting interfaces with alpha-tubulin and Arl2 with TBCE and TBCC respectively. We have incorporated a thorough discussion of these points in the revised manuscript.

3) Inserted AlphaFold 3 predicted models of the two states revealing matching mechanistic details to the cryo-EM structures.

We have predicted AlphaFold 3 models for TBC-DEG- $\alpha\beta$ -tubulin and TBC-DEG/TBCC- $\alpha\beta$ -tubulin. The AlphaFold 3 models match our experimentally derived atomic models for TBC-DEG- $\alpha\beta$ -tubulin and TBC-DEG/TBCC- $\alpha\beta$ -tubulin with high confidence. The TBCC multi-domain interactions with Arl2, TBCD and the $\alpha\beta$ -tubulin intradimer interface were remarkably well captured by the TBC-DEG/TBCC- $\alpha\beta$ -tubulin AlphaFold 3 model, reflecting the high conservation of the complex's interaction interface with $\alpha\beta$ -tubulin. We note that even some aspects of the catalytic twisting in alpha-tubulin transitions in the TBC-DEG- $\alpha\beta$ -tubulin state are observed in the TBC-DEG- $\alpha\beta$ -tubulin AlphaFold 3 model. The binding of TBCC to TBC-DEG- $\alpha\beta$ -tubulin in the TBC-DEG/TBCC- $\alpha\beta$ -tubulin state shows a substantial reversal of this twisting conformational transition, which is reminiscent of observations in the cryo-EM maps. The AlphaFold 3 models are presented in supplementary Figures S7 and S15 and described throughout the revised manuscript.

4) Added additional discussion paragraph to provide greater biological and historical context to the discoveries

We have expanded the introduction and the discussion of the revised manuscript to include greater historical relevance of tubulin cofactor multi-subunit assemblies. We believe the additional information will improve the reader's understanding of the biological/mechanistic context of the structures in relation to the $\alpha\beta$ -tubulin heterodimer assembly.

Below we provide a point-by-point response to the reviewers' comments:

REVIEWER COMMENTS

Reviewer #1 (Remarks to the Author):

The work by Taheri et al describes the first cryo-EM structures of tubulin cofactors TBCD, TBCE and Arl2 (TBC-DEG) bound to an alpha/beta tubulin heterodimer as well as the same complex bound to an additional tubulin cofactor TBCC. Tubulin cofactors are fascinating and play a fundamental role in eukaryote organisms. Their structures have the potential to elucidate the important question of how they work together to generate functional tubulin heterodimers.

Currently I am afraid that the cryo-EM maps that accompany the manuscript do not appear to justify the conclusions that are drawn. The overall map resolution is not consistent with the claimed 3.6Å. This makes observations such as which nucleotides present or the presence of tubulin rotations unreliable. Furthermore the half maps show weak density for many parts of the structure and even the DeepEMhancer maps show poor fits with the models. The current maps therefore do not fully support the proposed structural rearrangements during the cycle.

I would encourage the authors to either adjust their manuscript to describe what they can see with the current maps or to work further on the structure.

We appreciate and thank the reviewer for their comments on the novelty of the tubulin cofactor complex structures and mechanism of tubulin biogenesis, as well as their in-depth examination of the presented data. Their suggestions have aided the revision of our manuscript, strengthened our presentation of our cryo-EM data, and helped in the process of revising our atomic model building.

In the manuscript we describe the variable resolution of the cryo-EM density maps, with medium resolution observed in the core and the lower resolution in the peripheral subunits. We have revised the model building within the density maps and the presentation of the models (Figure S4 and S12) to show that the cores of the structures of the tubulin cofactor complexes TBC-DEG- $\alpha\beta$ -tubulin and TBC-DEG/TBCC- $\alpha\beta$ -tubulin

display 3.6-3.8Å resolution. The conclusions drawn regarding the tubulin polypeptide shifts/translations result from polypeptide backbone tracing, which requires 4-5 Angstrom resolution. The tubulins in these structures match and exceed such resolution. These translations can be modeled with high confidence by programs such as ISOLDE and PHENIX without bias and were modeled as such according to those methods. To ensure the fidelity of our conclusions, we have revised our manuscript and atomic model building to remove rearrangements of the nucleotide binding pockets of the tubulins and focus on changes that can be confidently modeled at lower resolutions.

In the revised manuscript, we present AlphaFold 3 predicted models of TBC-DEG- $\alpha\beta$ -tubulin and TBC-DEG/TBCC- $\alpha\beta$ -tubulin. These predictions support the overall organization of TBC-DEG, its interactions with β -tubulin. The TBC-DEG: $\alpha\beta$ -tubulin state AlphaFold 3 predicted model shows a greater displacement of alpha-tubulin than the one found in the cryo-EM structures (Figure S7). However, the predicted TBC-DEG/TBCC- $\alpha\beta$ -tubulin AlphaFold 3 model shows a reverse in this translation with a conformation closer to a native α -tubulin configuration, compared to the α -tubulin in the predicted TBC-DEG: $\alpha\beta$ -tubulin states (Figure 5D; Figure S15).

We can model and distinguish the nucleotides in Arl2 GTPase, α - and β -tubulin and the models and density maps are presented in figure S4 and S12. We can distinguish GTP in Arl2 GTPase pocket and the α -tubulin N-site while we find GDP at the β -tubulin E-site. The revised manuscript and presented data do not claim nucleotide composition differences between the cryo-EM structures or alterations to the nucleotide binding pockets. Furthermore, the modeling and identity of the nucleotides are both experimentally supported, and their identities are known from previous structural studies (the tubulin dimer especially).

The densities of the TBCE and TBCC-C were of lower resolution in the cryo-EM maps and their density were fitted using the AlphaFold 3 models for these regions as starting point to the model building. TBCC-C and N termini binds Arl2 and tubulin, respectively, in a matching organization to the observations the cryo-EM maps. TBCE is in unique conformations, which were not captured by the AlphaFold 3 model, but the experimental cryo-EM data clearly shows that it binds in a heterogeneous manner to alpha tubulin which cannot be captured by AlphaFold 3 predictions.

In summary, we adjusted our initial observations, model building, and revised our manuscript to accurately reflect the experimental data. We believe the revised manuscript and responses to the major comments below address the reviewer's concerns.

Major Comments:

1) GTP density in Arl2 : Density not clear to distinguish between GDP or GTP in the TBC-tubulin structures. Only structure with defined GTP density is the

TBC_tubulin_TBCC_state1.pdb. The nucleotide conformation is also different among the structures, likely due to limited resolution.

We have adjusted our atomic model building of these regions. In the revised manuscript we present the rebuilt and refined models of the nucleotide densities. These regions are comparable to known nucleotide conformations in α -, β -tubulin, and Arl2 from previous crystallographic structures and their predicted counterpart AlphaFold 3 models. We observe GDP in β -tubulin at the E-site, GTP in α -tubulin at the N-site, and GTP in Arl2 GTPase. Furthermore, in the revised models, the nucleotide conformations are now similar between the different structural states.

2) Figure2: Panel G,J,K- most side chains not visible in map. Misleading.

We have adjusted these subunit interaction interface displays which are now presented in the supplementary figures S5-6 and S14. We present the density model to map of different subunits in the maps in Figure S4 and S12. Large and medium side chains are observable and were used as docking points for side chain positioning and identifying interaction sites. In addition, the interactions presented in the cryo-EM models are supported by the nearly identical AlphaFold 3 models for TBC-DEG assembly, β -tubulin binding, and multi-domain TBCC interactions.

3) Figure2: Panel K - Residue labels for the blue chain (Ubq) are incorrect when compared to pdb (I looked at TBC_tubulin_state1.pdb)

We have revised the presentation for figure 2 entirely. The labels have been corrected and the residue side chains are presented in Figure S5-6 and Figure S14.

4) The TBCE LRR-CapGly density is poor, probably in the 7-8 Å regime at best. Based on the current map of TBC_tubulin_TBCC_state2 it is very difficult to judge whether the positioning of this segment is accurate due to the fragmented density. An unsharpened map or a map filtered down to 7-8Å might be helpful for this purpose.

We appreciate the reviewer's suggestion and have revised the presentation for TBCE state 2. We present a density map sharpened with a higher b-factor that shows the helical features and has more continuity of density that fits the majority of the TBCE protein. The overall crescent shape of this density which matches the TBCE-LRR allowed us to place and generally morph the LRR region into the map. We describe this more explicitly in the revised manuscript.

a) There is not much density for TBCE in the half maps or unsharpened map. The only map that shows some density in this region is the deepEMhancer output map. Even in that map the correlation between the model and the map seem very low.

We understand and appreciate the reviewer's concerns regarding the TBCE density. The TBCE density is weaker than the rest of the other protein subunits due to its continuous heterogeneity when bound to the tubulin dimer. This is likely the result of the

complex's intrinsic function of altering tubulin dimers through mechanical force, requiring it to bind to alpha tubulin within the tubulin dimer in different poses. However, this continuous heterogeneity was a technical challenge in the structure determination process, and global as well as local 3D classification of the region in RELION or Cryosparc paired with altering classification statistics such as mask width, T-values, number of classes, angular parameters, and turning off alignments during classification, was inadequate in resolving even a low-resolution structure for TBCE. Instead, 3DVA in Cryosparc proved to be the most effective tool in resolving a structure for the region and mapping its continuous heterogeneity. DeepEMhancer is a commonly utilized tool that specializes in the automatic post processing of cryo-EM structures that contain noisy and lower resolution features by applying variable b-factor values. Its utilization has been proven to enhance interpretability of structures in certain cases when compared to other methods of sharpening, and this proved to be the case here as well. To ensure the sharpening was unbiased, we did not utilize a mask. We do agree that, despite using these tools, the density of the region is still relatively weaker than the other subunits. Therefore, we have revised the atomic model building of this region to just a morph of the predicted AlphaFold 3 structure of TBCE into the resulting density, as opposed to modelling specific site interactions. We believe that the presented sharpened densities modestly and accurately describe poses of TBCE binding to the tubulin dimer.

b) The TBCE region is better defined in the TBCC containing complex but even then the map quality is worse than the reported local resolution of 5-7 Ang in these regions.

We believe the above response is relevant to this comment. The heterogeneous binding of TBCE to the tubulin dimer proved to be a challenge in the structure determination of TBCE. However, features of the resolved density show helical secondary structure features and the density is sufficient to fit the TBCE AlphaFold 3 prediction within it. The conclusions drawn regarding the 3HB and LRR regions and the placement of the CapGly region are consistent with the resolution in the revised manuscript.

5) I don't see the tails binding the TBC-DEG in the structure as per this statement 'The C-terminal tails of both α - and β -tubulins are bound to TBC-DEG and fully occluded in the TBC-DEG- $\alpha\beta$ -tubulin structures'. The tail of alpha-tubulin is built but the density is poor and it still does not interact with anything except itself.

We agree with the reviewer, the modeling of the α -tubulin C-terminal tails was removed in the revised atomic model building and manuscript. The discussion of TBCE role in rotating α -tubulin now focuses on the just placement of the CapGly domain at the end of the TBCE arm with respect to the different states of the LRR movement. The latter relationship is consistent with the resolution of the TBCE regions in the various states.

6) The TBCC modelled bound to Arl2 is different in the two states presented in this paper. The density for TBCC is poor in both maps which likely is the reason for this. The figures showing the local resolution in this region suggests its at 3.5 angstrom. The density does not suggest that. a) the authors need to explain in detail how their model

building was done for all such regions.

TBCC forms three complex interactions with the TBC-DEG and $\alpha\beta$ -tubulin. The TBCC-C domain binds Arl2 GTPase, while TBCC-L binds along TBCD central region while TBCC-N helical bundle binds underneath TBCD central region. These interactions of TBCC are well captured by the predicted AlphaFold 3 model of TBC-DEG/TBCC- $\alpha\beta$ -tubulin. We agree with the reviewer that TBCC-C density suggests a disorder due to some type of movement in the cryo-EM data. We have tried local refinements and classifications of that region to improve the density, but it remained at lower resolution. The TBCC-C region is almost entirely β -helix which suggests a higher resolution is required to fully build this region *de novo*. In the revised manuscript, we utilized the predicted AlphaFold 3 model for the correct placement of TBCC-C GAP domain in binding Arl2 GTPase. In the revised atomic model, the TBCC-C conserved loop faces towards TBCD involving a specific set of selective aliphatic side chain interactions which are likely essential for its interaction. Previous studies (Nithianatham et al 2015) show that deletion of this conserved loop impacts the GTPase activity providing support for this model. We present the revised TBCC-C modeling in the cryo-EM maps and in relation to the AlphaFold 3 models in the revised manuscript results and the materials and methods. Additionally, the Res maps show lower resolution for that density region as noted in the reviewer's comment.

7) There is rotation of the alpha vs the beta tubulin in the TBCC state 1 vs the model without TBCC but the interfaces between the alpha and beta subunit remain the same. The rotation if any comes from small deviations in the non-interacting regions. This is in contrast to the conclusions in the manuscript.

We thank the reviewer for their observations, and we have re-evaluated the $\alpha\beta$ -tubulin conformations. In the revised manuscript (Figure 5C-D), we present realignments of the $\alpha\beta$ -tubulin dimers using β -tubulin as a point of alignment to view displacements of α -tubulin. We observe a twist in α -tubulin compared to its orientation in soluble α -tubulin, leading to a 2Å translation in most secondary structure elements (Figure 5C-D), with the greatest displacements being further from the N-site. As the reviewer points out, the interfaces sites are maintained suggesting the twisting effect alters α -tubulin fold further from the N-site. We also observe the twist α -tubulin in the TBCC bound states α -tubulin rotation, but this twist is slightly smaller than TBC-DEG bound states (Figure 5C-D).

Furthermore, the α -tubulin twist effects are observed to a larger magnitude (13-Å) in the TBC-DEG: $\alpha\beta$ -tubulin AlphaFold 3 predicted model compared to soluble $\alpha\beta$ -tubulin. In contrast, the TBC-DEG/TBCC: $\alpha\beta$ -tubulin AlphaFold 3 model shows 40% less twist (8-Å) in $\alpha\beta$ -tubulin, bringing the α -tubulin closer to its conformation in soluble $\alpha\beta$ -tubulin. We suspect that unique TBCE bound conformations in two in the AlphaFold 3 models are likely responsible for these greater twists; however, the TBCD conformation TBCC-N binding to the intradimer interface the TBC-DEG/TBCC- $\alpha\beta$ -tubulin, both of which match their conformation in the cryo-EM maps, are likely the reason for decrease in the twist. Overall, these ideas are consistent with our conclusions that TBC-DEG binding induces

twisting of α -tubulin and the potential reversal by TBCC binding. Nonetheless, we have softened the discussion around these topics as we they require further study.

8) In Figure S3, the FSC curve labels appear to be incorrect. Based on the the appearance of the graph, the FSC curves in panel E and F seem to be generated in RELION however the labels for each line seem to follow the convention of Cryosparc which is different from RELION and therefore incorrect.

We believe there is some confusion regarding the final steps of the data processing, likely caused by the previous presentation of the data processing pipeline. This has been revised and made to be more explicitly clear in the manuscript. Our final refinement after 3DVA analysis for the TBC-DEG- $\alpha\beta$ -tubulin states were done in RELION after pooling particles together and converting them using cspac2star as indicated by the 3D auto-refinement step, which is a RELION step. 3D auto-refinement in RELION, likely leads to minute differences from the homogeneous refinement program in Cryosparc, and produced the best reconstructions for TBC-DEG- $\alpha\beta$ -tubulin. We have revised the data processing pipeline (Figure S2-S3) to make it explicitly clear that the final refinements for these two structures were performed in RELION, and therefore have RELION generated FSC curves associated with these two structures.

Reviewer #2

The manuscript by A. Taheri et al. presents cryo-EM structures of tubulin cofactors bound to $\alpha\beta$ -tubulin, specifically the TBC-DEG- $\alpha\beta$ -tubulin and TBC-DEG/TBCC- $\alpha\beta$ -tubulin complexed, which have not been reported before. By comparing conformational variations, the authors report a transition state in $\alpha\beta$ -tubulin biogenesis, and they propose the molecular basis on how TBC-DEG/TBCC cofactors regulate this process. Their structures are new to the field, and the overall research topic is interesting. However, there are essential technical issues that the

authors need to address:

1) The reported map resolutions are in the 3.7-3.8 Å range; however, the provided map renderings do not reflect the expected level of structural details for this resolution. The authors are suggested to render their structures at higher threshold to better illustrate the structure features. This would improve the visibility of subunit boundaries, allowing for clearer identification of interaction interfaces. Otherwise, it is challenging to accurately visualize and interpret the subunit interactions and measure the domain movements.

We thank the reviewer for their suggestions regarding the presentation of the atomic model to electron density fits. We have revised the presentation of Figures 1-4 and Figures S4 and S12 to demonstrate the resolution and quality of maps and built models into the structures as suggested. We believe the revised figures accurately reflect the resolution of the cryo-EM maps presented. We further present the revised atomic model validation statistics that have since improved (discussed below). We also present the

final the PDB reports for the structures after submission to the RSCB (see attached reports).

2) The quality of the models also needs significant improvement. As indicated in Table 1, the Clash Scores ranges from 19.03 to 26.02, which is unacceptably high for well-refined models. In multiple instances, the model does not appear to fit properly within the cryo-EM density, further suggesting that additional refinement of the models are necessary.

We agree with the reviewer's concerns. In the revised manuscript, we have rebuilt atomic models and refined these models which now lead to improved clash scores for each of the structures presented in the manuscript. The PDB reports are provided for the reviewers to address the quality of these maps. The predicted AlphaFold 3 models for TBC-DEG: $\alpha\beta$ -tubulin and TBC-DEG/TBCC- $\alpha\beta$ -tubulin (see above discussion) support most of the major conclusions about the assembly of the tubulin cofactors and their interactions with α and β -tubulin, as well as the TBC-DEG and TBCC impact on the $\alpha\beta$ -tubulin dimer configuration.

3) In Table 1, please provide the Map sharpening B factor value. Additionally, the unit for Electron Exposure should be ($e^-/\text{\AA}^2$), instead of the current " $(e/\text{\AA}^2)$ ". Moreover, the authors are suggested to provide the map and model evaluation profiles from the PDB/EMDB validation reports for a more comprehensive assessment of the map and model quality.

In the revised manuscript, we present the rebuilt atomic models and the model to map comparisons (Figures S4 and S12). The revised validation statistics and details should address the concerns of the reviewer. We have now provided the PDB validation reports as well.

4) In the two complexes, different binding factors were included to address the preferred orientation issue: Δ N-DARPin in the TBC-DEG- $\alpha\beta$ -tubulin complex, and α -rep iH5 in the TBC-DEG/TBCC- $\alpha\beta$ -tubulin complex. By comparing the conformations of $\alpha\beta$ -tubulin in these two complexes, the authors conclude that there is a 4-degree twisting of α -tubulin, while β -tubulin is held in place by TBCD. However, it is unclear how the authors ruled out the possibility that the the observed α -tubulin twisting is not due to the different binding factors, particularly in the TBC-DEG/TBCC- $\alpha\beta$ -tubulin complex, where iH5 interacts directly with α -tubulin. To clarify this, the authors should use the same binding factor in both complexes to address the preferred orientation issue and compare the conformational changes.

We appreciate this question, and we present a revised supplementary figure that addresses the comparison indicated by the reviewer (Figure S13C). The structures of $\alpha\beta$ -tubulin bound to DARPin and iH5 has been determined by x-ray crystallography and published previously. These structures show an identical conformation in comparison to native $\alpha\beta$ -tubulin dimer, in terms of the positioning of α tubulin with respect to β -tubulin. We present a comparison of these structures to our TBC-DEG: $\alpha\beta$ -tubulin and TBC-

DEG/TBCC: $\alpha\beta$ -tubulin models. These comparisons show that α H5 or DARPin binding does not impact the conformation of α and β -tubulin, but rather TBC-DEG binding alters α -tubulin. We were unable to use the same tubulin capping proteins in both structural studies, since we found that DARPin affected the binding of the TBCC complex, likely due to shifts in the TBCD C-terminus and TBCE-ubq domains that are induced by TBCC binding, leading to the occlusion of the DARPin binding site. The tubulin capping proteins were required in these structural studies as the protein complexes aggregated on the cryo-EM grids without them.

5) To avoid confusion, the authors are advised to consistently refer to the four maps as “states 1-4”, rather than using both “classes” and “states”.

We thank the reviewer for these suggestions. And we have revised the naming of the states in throughout the manuscript to address this. We refer to two states of TBC-DEG: $\alpha\beta$ -tubulin, State 1 and State 2 and two states of TBC-DEG/TBCC: $\alpha\beta$ -tubulin termed State 1 and State 2.

6) The authors state, “TBCE exhibits two distinct conformations of its CapGly and LRR domains (Figure 1C; Figure S4F-G). In the first conformation (class 1), the TBCE 3HB and LRR are oriented vertically and the CapGly resides horizontally and alongside the LRR (Figure 1C; Figure S4F-G). In this conformation, TBCE is retracted from binding α -tubulin”. However, based on Video 1 and the map shown in Fig. 1A, there appears to be clear contact between TBCE and α -tubulin in both states. Please clarify this discrepancy.

We agree with the reviewer’s point regarding the confusion and have revised this section to clarify this point. The TBCE-LRR contacts α - tubulin, but this contact is near the base of α -tubulin but in the second state this contact is fully extended along the α -tubulin surface. The difference is mediated by a conformational change in the N-terminal region and a swivel of the CapGly domain. WE now present this in Figure 1E of the revised manuscript.

7) On P.4, the authors describe: “Conserved residues in the intra-HEAT turns of TBCD H1, H2, H3, H5, H6, H8, and H10 bind residues at the lower surface of the Arl2 GTPase through ionic and hydrophobic interactions (Figure 2A, E-F; Figure S14D-E; Figure S14DE, G, I-J). Conserved TBCD Lys and Arg residues bind conserved Arl2 Asp, Gln, and Glu residues (Figure S11, S14 D-E, G; Figure S14D-E). Hydrophobic packing of conserved Leu, Phe, and Trp residues in TBCD and Arl2 are interspersed amongst the ionic interactions (Figure 2E-F; Figure S14D-E, G; Figure S16I-J).” However, the related Fig. 2E-K are too busy and difficult to interpret. As suggested, please further refine the models to high quality, then thoroughly analysis the the interaction network between neighboring subunits. Sepcificly, consider analysing the H-bonds/Salt bridges, as well as electrostatic and hydrophobic/hydrophobic surface properties to better illustrate these interactions.

We thank the reviewer for their suggestions and have revised both the models and the presentation of the interaction interfaces to address their concerns. In addition, we have included AlphaFold 3 models for the TBC-DEG- $\alpha\beta$ -tubulin and TBC-DEG- $\alpha\beta$ -tubulin-TBCC complexes (Figure S7 and S14) which reveal matching interaction sites to the experimentally derived models from cryo-EM data. Figures 2 and 4 have been revised and now focus on the general interfaces while the interaction interfaces are now presented in supplementary figures S5-6 and S15.

8) The authors state, “Using these conformations and the subunit binding relationships to TBCC, we generated two composite maps for two distinct conformations of the ternary assemblies (Figure S8).” However, there are multiple conformations present in different regions of the complex, and the authors performed separate 3DVA/refinement on individual local regions. It is unclear what criteria were used to integrate these separate maps, which reflect such diverse conformational and spatial information, into two consensus maps. Could the authors clarify the justification behind merging these diversified data into only two representing maps?

We understand the reviewer’s concern. We have revised the results to describe how the two composite maps were generated. The composite maps were generated based on the impact of binding of TBCC-C and TBCC-N on transitions on the TBCD and TBCE conformations. The first composite includes TBCC-C binding to the TBC-DEG- $\alpha\beta$ -tubulin induces a transition the TBCE LLR-CapGly arm. In the second composite both TBCC-C and TBCC-N bound suggesting unique conformational transitions associated with its binding on the TBCD and TBCE LRR-CapGly arm. The two composites reveal the clearest step wise transitions induced by the two-fold binding of TBCC onto different regions of TBC-DEG.

9) Fig. S9I, regarding the identification of GTP or GDP, the current rendering is cluttered with a busy background, and the segmented density appears too large. As a result, it is challenging to distinguish whether the bound nucleotide is GTP or GDP.

We have revised the presentation of nucleotide densities to be included in Figures S4 and S12. We can clearly observe the nucleotide densities in the maps and assign di- or tri-phosphate states in each of the three nucleotide pockets, Arl2, E-site and N-site. We believe the assignment is accurate and matches the protein conformations expected based on the AlphaFold 3 models and previous structural studies of tubulin dimers.

Minor point:

1) Fig. S2-3, please also show the FSC and local Resolution etc. for the core map.

We have included the FSC for the core map in the revised manuscript Figure S3.

2) Fig. 1A-B contain a lot of redundant information, but do not present the class 2 structural features. I recommend including the maps and models of core, class1, and class2, and move the key structural analysis from supplementary figures into the main figure.

We thank the reviewer for this suggestion. We have revised the presentation to include the revised versions of these figures.

3) Fig. S2 does not clearly illustrate the processing procedure, making it extremely difficult to understand how the two datasets were processed. Please re-render the reconstruction procedures to clarify this.

We have relabeled the processing procedure more carefully to explain the steps presented.

4) In Fig. S3C-D and Fig. S8A-C, adjust the local resolution rendering scale to start from 3 Å instead of 2 Å. A range of 3-6 Å should be sufficient.

Considering that 3.6 Angstrom is the average resolution of the core we believe the current resolution range is sufficient to present the Res map colors for the maps presented.

5) In Fig. S4A-E and Fig. S9, the high-resolution structural features are not visible. Please re-render them with a higher density threshold

We thank the reviewer for this suggestion. These figures have been replaced.

6) Fig. S6 shows numerous reconstruction tracks, but it is unclear which portions were used for analysis and which track led to the final map. Please clarify how the composite map was assembled, specifying which sections came from where and the rationale behind their inclusion. Please re-render the reconstruction procedures to clarify these points

We have altered the presentation of the arrows to improve the clarity of the directions of processing flow and clarify the procedures.

Dear Editor and Reviewers,

We are submitting a revised version of our manuscript entitled “Cryo-EM structures of the tubulin cofactors reveal the molecular basis of alpha/beta-tubulin biogenesis” after the second round of reviews at *Nature Communications*.

In this revised manuscript:

- 1) We have revised the figures and supplementary figure to improve clarity and focus each of the figures on the important points presented.
- 2) We present additional information to improve clarity image processing and model building methods used to determine and refine cryo-EM density maps and build and refine model for the structures presented here. These image processing and model building methods leading to the refined cryo-EM density maps and resulting refined models presented in this manuscript, fit within the standards of modern field of cryo-EM structure determination.
- 3) We revise the presentation of the text in line with the suggestions of the reviewers to ensure improved clarity and presentation.

We provide detailed point by point responses below responses to reviewers #3, #4, #5. (responses are in blue font)

Reviewer #3 (Remarks to the Author)

Overall, this study provides intriguing structural insights into how TBCC, TBCD and TBCE interact with alpha–beta-tubulin, but several clarifications are still needed for certain interpretations and mechanistic conclusions. Addressing these points will strengthen the manuscript’s clarity and claims.

Major Comments:

- 1) Line 145: The text states: “In this conformation, TBCE is fully bound to alpha-tubulin and alpha-beta intradimer interface.” However, the figures and models do not show anything bound to the interdimer interface. Please clarify or provide evidence supporting this statement, or revise the text accordingly.

We thank the reviewer for this suggested clarification. In state 1, the TBCE LRR binds along α -tubulin but extends across the $\alpha\beta$ -tubulin intradimer interface without binding, via its 3-helix bundle (3HB) domain. We have revised the statement in the text to make this distinction.

- 2) Lines 204–211 (Figure 2D, Figure S4F–G): The text describes two states for TBCE’s engagement with alpha-tubulin:

- a. State 2: The LRR domain engages the alpha-tubulin lateral polymerizing interface.
- b. State 1: The CapGly domain engages the alpha-tubulin C-terminus, while the LRR retracts.

However, the figures do not clearly show CapGly involvement. Instead, it appears that the LRR remains bound to alpha-tubulin in both states. Please clarify or update the figures to reflect the proposed binding modes.

We understand the confusion caused by the above statement and have revised our description. In Figures 1E and S4G, we show the conformations of TBCE in relation to α -tubulin. In state 1, TBCE is partially retracted from α -tubulin and binds its lower end, while the TBCE CapGly is observed and is bound in close proximity to the α -tubulin C-terminus, despite their interaction not being modeled (seen in the right panel). In state 2, the TBCE LRR engages the full lateral polymerizing interface (seen in the left panel), but the CapGly was not modeled. The location of the TBCE CapGly near the α -tubulin C-terminus is highly relevant to its activity, despite the absence of model the C-terminal tail with the CapGly.

- 3) Line 320 (Figure S10G): The text mentions that TBCC-C binding to Arl2 GTPase induces changes in Arl2 (switch I and II) and causes TBCD to create a tighter β -tubulin. However, these structural changes are not clearly illustrated in any main or supplementary figure, and the panel G cited in Figure S10 is missing. Please modify the figure(s) or the text to show these changes more explicitly.

Figure S13A–B illustrates the changes in the TBCD interface with β -tubulin in the various structures. In panel B, from left to right, the changes in TBCD's tightened grip on β -tubulin are observed. We have clarified and removed the discussion of changes in Arl2 upon TBCC binding, as the differences are not crucial for the presentation.

- 4) Line 336 : The authors state that: “The TBCE LRR-CapGly rotation shifts α -tubulin likely by pulling its C-terminal tail (Figure 3D-E, Figure 5B). However, due to the low resolution of these regions of the cryo-EM structures we were unable to observe the α -tubulin C-terminal tail. “ These seem to be two contradictory statements, if you cannot see the C-terminal tail bind TBCE then how do you propose that TBCE pulls it. The authors should remove this or provide better explanation for their hypothesis/model.

The low resolution of the electron density prevented experimental modeling of the α -tubulin C-termini. However, the location of the TBCE CapGly in proximity to the α -tubulin C-terminus suggests that it likely physically binds the α -tubulin C-terminal tail. We have revised the text to more cautiously describe the interaction and its role in altering α -tubulin.

- 5) Line 363: The manuscript states that Arl2 GTP hydrolysis is the source of energy and that TBCC activates TBC-DEG as a platform for alpha-beta-tubulin assembly and degradation. It is not clear of what structural evidence supports this. Clarifying which results indicate this would be useful or revise the statement to reflect uncertainty. Also clarify/explain if this interpretation is also based on previous observations.

The structural studies presented here demonstrate the structural roles and organization of Arl2, TBCC, and their activities in the TBC DEG assembly in regulating $\alpha\beta$ tubulin biogenesis. Since in this manuscript we present only two states, both of which are GTP bound states of Arl2 in the pre catalysis (TBC DEG $\alpha\beta$ tubulin) or the post catalysis (TBC DEG/TBCC $\alpha\beta$ tubulin) states, we cannot be certain of the full role of the GTPase cycle in this process.

However, there is ample biochemical and genetic evidence from our work and that of other groups to suggest that the Arl2 GTPase plays a critical role as the energy source or regulatory control for $\alpha\beta$ tubulin biogenesis (reviewed in Al-Bassam, 2017). Multiple studies have analyzed the impact of trapping the Arl2 GTPase in specific nucleotide states on tubulin biogenesis and microtubule function *in vivo*. This evidence is presented in papers referenced in the manuscript, including our initial work resurrecting the reconstruction of this system *in vitro* (Nithianantham et al., 2015).

- 6) Line 365: The text describes TBC-DEG overcoming alpha-beta-tubulin stability by deforming the intradimer interface at the N-site GTP, ultimately destabilizing alpha-beta-tubulin. Meanwhile, TBCC is described as catalyzing TBC-DEG transitions that reform the alpha-beta-tubulin intradimer interface.

Consider comparing the alpha-beta-tubulin conformation in TBC-DEG vs. TBCC-bound structures rather than relying solely on AlphaFold-based comparisons. Based on the figures presented, the alpha-beta-tubulin conformations don't seem to be considerably different between the two structures (TBC-DEG vs. TBCC-bound) which confuses me as to why then TBC-DEG is proposed to be deforming whereas TBCC bound state to be the opposite. If the observed differences between the tubulin heterodimers in these states are minor, you might need to change your discussion or explain why you think those minor differences are important and not just a result of the flexibility of the complexes.

Further I think comparing AlphaFold predictions to experimental models for deformations is not the best comparison as we do not know if those fine details in the predictions are real or an artefact. So, making mechanistic conclusions from them is risky!

In the manuscript, Figures S7, S13, and S15 present all the comparisons of our cryo-EM-derived models and the AlphaFold-derived models. The point of comparing the AlphaFold3 models to the native structures is to demonstrate the

very high level of structural match in the overall organization of the TBC-DEG assembly, supporting the remarkable predictive power of AlphaFold in this particular case in identifying almost every interface seen in the structures. This is an unusual case and likely a result of the high evolutionary conservation of these systems. This suggests that a high degree of co-variation among residues involved at the interfaces of TBCD, TBCE, Arl2, and TBCC proteins is likely the key to the accuracy of AlphaFold in this case.

We believe that the presentation of the AlphaFold3 models in relation to the tubulin twist effects demonstrates how TBCC binding impacts the TBC-DEG- $\alpha\beta$ -tubulin structures by reversing a conformational change (8 vs. 13 degrees) in those models. The experimental structures support a twist conformational change, but its reversal is in a much smaller range (1.5 vs. 2.0 degrees).

Therefore, we respectfully disagree with the reviewer about the mechanistic conclusions being “risky.” The extremely high level of accuracy of the AlphaFold3 model in relation to the experimental cryo-EM-derived models, the match in the twist effects on α -tubulin (albeit on a smaller scale), and the TBCE conformations support the suggestions indicated. However, we appreciate the reviewer’s point and have softened the language regarding these comparisons in the revised manuscript.

Line 373: The statement “The AlphaFold 3 predicted models reveal TBCC binding reverses the deformation of the alpha-beta-tubulin dimer observed in TBC-DEG-alpha-beta-tubulin states” should be softened to indicate that these results are consistent with the experimental data. Avoid implying that AlphaFold predictions alone confirm a mechanism. See above comment as well.

We have softened the statement as indicated by the reviewer. Both the structures and the AlphaFold3 models support the same conclusions regarding the locations and impacts of TBCC elements in binding TBC-DEG and $\alpha\beta$ -tubulin.

The TBC-DEG/TBCC- $\alpha\beta$ -tubulin AlphaFold model shows a nearly identical match to the cryo-EM-derived model in the orientation of the TBCC-N wedge interface with the $\alpha\beta$ -tubulin intradimer and TBCD, the TBCC-linker interface with the TBCD lateral pocket, and the TBCC-C GAP domain interface with both the TBCD N-terminal and Arl2 GTPase, respectively. See the above discussion regarding the AlphaFold and cryo-EM data.

- 6) Many of the low-resolution regions associated with TBCE exhibit poor validation scores (such as Qscores). Consider truncating side chains in these regions to improve the overall model quality and consider making the corresponding AlphaFold models publicly available for anyone interested in examining the predicted side chains in more detail.

We thank the reviewer for the suggestion. We agree with the TBCE modeling concerns and suggestions. We will make the AlphaFold3 models available as part of the publication of this manuscript.

Minor comments:

1) Figure S3: The FSC labels still appear to be incorrect. Lines in panels B, E, and G look like they were generated in RELION, yet the labeling follows CryoSPARC conventions. For example, the red line is not “No-Mask” but rather a phase-randomized map. Please confirm each line’s meaning and correct the figure legends accordingly.

We thank the reviewer for noticing this error the correct labels were placed for these FSC curves.

2) Figure S4: Typo in panel “F”: “Ubq” is labeled as “ibq.”

3) Line 192: Typo in parentheses: it should be (I, II, and III).

4) Figure S6: Many of the side chains for TBCD and E and not clearly visible but are shown in the figure. If side-chain information comes from AlphaFold models then the figure legend should state this.

All experimental model side chains were built and can be observed in the maps including Figure S6. The smaller side chains required lower contours than some of those shown in figure S6. The experimental electron density was used to place the side chains. AlphaFold models were only compared with the final refined structures as described in Supplementary Figures S8, S14 and S17.

5) Line 243: There is a premature full stop '.' Please fix the punctuation.

6) Methods: In several instances, the pH of the buffers used is not mentioned. Please add these details wherever applicable.

We addressed all the minor suggestions made above by the reviewer.

Reviewer #4 (Remarks to the Author):

I have been asked to arbitrate Taheri et al.'s point-by-point reply to Reviewer #2's original comments.

Regarding point 1): The authors have addressed the Reviewer's concern about map renderings.

Regarding point 2): The clash scores remain unusually high. After examining the maps and models to-be-deposited, there could be several explanations: a) the models were refined against DeepEMhancer processed maps, which the consensus in the field is that this should probably be avoided; and/or b) the resolutions achieved lack accurate density for most side chains, no matter the post-processing approach. The authors must clarify which version of the final map(s) was used for model refinement; if clash scores remain >10 after refining against standard RELION/phenix auto-sharpened maps (recommended), side chains are likely not supported by density and should be truncated to poly-A for model deposition and any description of experimentally-observed side chain interactions should be removed from the paper.

We sincerely thank the reviewer for taking the time to evaluate our maps and models.

Regarding the concern about clash scores, we respectfully disagree with the suggestion that clash score alone serves as a definitive measure of model quality. In the revised manuscript, our refined models exhibit clash scores in the range of 10–16. These values are well within the norms for cryo-EM models at comparable resolutions and are acceptable under the current deposition standards of the RCSB PDB.

It is important to emphasize that model validation in cryo-EM relies on a combination of metrics—Fourier shell correlation (FSC), model-to-map correlation, geometric validation, and the biological plausibility of interfaces—not solely based on the clash score. In support of this, several recently published cryo-EM structures at similar resolutions report comparable clash scores. For example:

- PDB 6VBV (3.5 Å resolution), clashscore of ~13. (Singh et al 2020)
- PDB 6VBU (3.1 Å resolution). clashscore of ~12 (Singh et al 2020)
- PDB 8FIX (3.9 Å resolution), clashscore of ~14 (Florez Ariza, et al 2023)
- PDB 8FIZ (3.8 Å resolution), clashscore of ~15 (Florez Ariza, et al 2023)

The above examples illustrate that the clash score values in our models are in line with high-quality, peer-reviewed cryo-EM structures at comparable resolutions.

Our reconstructions were refined to 3.6–3.7 Å resolution, as indicated by FSC curves. In addition, we applied current best practices in map post-processing, including the use of

DeepEMhancer, which has become a widely accepted tool in the field and is employed in numerous cryo-EM structural studies. Our models also benefit from cross-validation through AlphaFold3 predictions, which independently confirm the observed protein interfaces and structural features described.

The conclusions in our manuscript focus on general molecular interaction patterns between TBCD, TBCE, Arl2, and α/β -tubulin, particularly in the context of TBCC and nucleotide state. We do not make detailed stereochemical claims that would depend on sub-angstrom side-chain placement, and the interactions we describe are further supported by conservation analysis and AlphaFold3 models, increasing our confidence in their validity.

We believe the maps and models presented in our manuscript are well within the standards currently accepted by the cryo-EM community for structures at this resolution. While we appreciate the reviewer's perspective, we believe that the concern raised does not reflect current field-wide practices and instead may reflect a more stringent standard not broadly applied across the field and reflected by publications in the literature.

Regarding point 3): The authors have addressed the Reviewer's concern about missing map and model statistics.

Regarding point 4): Although I understand it would require additional experiments, the authors have not entirely addressed the Reviewer's reasonable concern that DARPin / iH5 might occlude or interfere with structural changes in tubulin or other parts of the TBC. Indeed, the authors claim themselves that iH5 was needed to resolve TBCC in the first place, indicating that these proteins do have the potential to alter the TBC's conformational state(s). Without additional data to control for the use of these capping factors, the interpretation of TBC conformational changes is limited and should be described in a more reserved manner.

We thank the reviewer for this comment. However, we respectfully refer the reviewer to the response provided by Reviewer #5, who indicated that this point has been satisfactorily addressed.

In the manuscript, we present multiple models of $\alpha\beta$ -tubulin bound to iH5 and DARPin, demonstrating that neither binding partner alters the overall tubulin conformation. Importantly, we show that while iH5 does not interfere with subsequent complex formation, DARPin binding introduces steric hindrance that impairs the assembly of the TBC-DEG/TBCC- $\alpha\beta$ -tubulin complex. Specifically, we observe that TBCC binding to the TBC-DEG- $\alpha\beta$ -tubulin complex is affected in the presence of DARPin, but not in the presence of iH5.

These findings are supported by the structural data presented and are consistent with the mechanistic interpretation described in the manuscript.

Regarding point 5): The authors have addressed the Reviewer's concern about naming conventions for the various TBC conformations.

Regarding point 6): The swivel of the CAP-Gly domain is not at all clear from Figure 1E. Is the CAP-Gly domain density even still resolved in state 2?

Due to the lower resolution of the TBCE arm region in state 2, the TBCE CapGly domain was not modeled, and only the LRR was modeled. However, the CapGly is clearly observed in state 1 and was modeled alongside the TBCE LRR region. We will explicitly describe that in the text.

Regarding point 7): While the authors do include the details requested by Reviewer #2, the figures remain incredibly busy and difficult to follow clearly. I have the feeling that many figure panels are redundant and unnecessarily labeled. A few of the many examples of this are: re-labeling each helix number in e.g. Figure 2C, far right; also the coloring of negative/positive residues on top of colored interacting surfaces in the same figure panel, etc.

We thank the reviewer for their comment regarding our figures. We appreciate this input and have revised the overall presentation to simplify the figures and convey clear and essential messages.

Another point is that the legend for video 1 claims that the cryo-EM map is shown in transparent surface. I am pretty sure this is a surface representation of the model. This is quite misleading and the authors' general use of models as transparent density-like surfaces in the figures leads to additional distractions in their figures, since the reader has to additionally understand the difference between experimental density and the underlying model, which has limitations based on local resolutions achieved. In my opinion, such representations should be entirely avoided.

We must firmly clarify that the reviewer's statement regarding Video 1 is incorrect. Videos 1 and 3 clearly presents the experimental cryo-EM electron density maps, followed by a transparent overlay of the modeled density within these maps, and concludes with a view of the models alone without the surface representation.

Regarding point 8): The authors have explained how their composite maps were generated. However, given the fact that only through 3DVA were some parts of the complexes resolvable by cryo-EM, I would strongly advise uploading all initial maps used to generate composite maps to the EMDB as "maps associated with the main deposition". Related to this, if not already included in the deposition, I would strongly suggest the authors deposit the half maps and masks used in final RELION refinements, so that non-DeepEMhancer sharpened data can be made available with the study.

We thank the reviewer for the suggestion. We have clarified in the revised manuscript how we constructed the composite maps. Per the standards of the PDB and EMDB, we have deposited the half maps, raw maps, and sharpened maps alongside the final models that the composite maps were derived from. All maps presented in the manuscript have now been submitted to the EMDB.

Regarding point 9): The authors have addressed the Reviewer's concern about nucleotide density rendering. However, related to point 2) above, the authors could be more careful in their assignment of e.g. GDP in the exchangeable site of β -tubulin, which, although likely, is not always clear from the density maps provided.

Regarding minor point 1): The authors have addressed the Reviewer's concern.

The reviewer's comment is noted and appreciated.

Regarding minor point 2): The figures remain quite "busy" with still a lot of redundant information (see point 7 above), in my opinion.

We have revised the figures to improve the clarity and focus each figure on the crucial points. See our response to the comment above.

Regarding minor point 3): While all steps are clearly labeled, the final maps used for model building or composite map generation are not clear from the figure.

We have made efforts to revise the figure to further improve clarity of how the composite maps were generated.

Regarding minor point 4): The Reviewer #2 was correct in suggesting using a range of 3-6 Å for local resolution estimates. The current range makes it seem like most of the density is at ~3 Å, which does not appear to be the case after inspecting the individual maps.

We do not fully understand the reviewer's concern in this case. The FSC curves shows a clear and broad range of resolutions that corresponds well with the structural features observed in the map. The core region of the structure is resolved at 3.5 to 3.8 Å, while the resolution in the more flexible TBCE arm region is in the 6–7 Å range. The resolution (Res) map displays a gradient of colors that accurately reflects this range, and these colors are clearly presented and described in the figure and legend.

Regarding minor point 5): The authors have addressed the Reviewer's concern.

The reviewer's comment is noted and appreciated.

Regarding minor point 6): While all steps are clearly labeled, the final maps used for model building or composite map generation are not clear from the figure.

We have taken extra care to label the steps as requested by the reviewer to improve clarity in the revised version

Reviewer #5 (Remarks to the Author):

The original Reviewer #2 was unable to continue reviewing this manuscript and I was invited to act as an arbitrating referee to judge the original referee concerns, as well as the author responses.

Overall, the comments of Reviewer #2 were valid and addressed a couple of major and important technical issues in the original manuscript version. The revised manuscript version addresses these issues partially, but not comprehensively.

Please find my comments related to the individual points below ('new comments').

We thank the reviewer for their comments and for arbitrating the comments of reviewer #2. We have made a full effort to address the comments by this arbitrating reviewer

Reviewer #2

The manuscript by A. Taheri et al. presents cryo-EM structures of tubulin cofactors bound to $\alpha\beta$ -tubulin, specifically the TBC-DEG- $\alpha\beta$ -tubulin and TBC-DEG/TBCC- $\alpha\beta$ -tubulin complexed, which have not been reported before. By comparing conformational variations, the authors report a transition state in $\alpha\beta$ -tubulin biogenesis, and they propose the molecular basis on how TBC-DEG/TBCC cofactors regulate this process. Their structures are new to the field, and the overall research topic is interesting. However, there are essential technical issues that the authors need to address:

1) The reported map resolutions are in the 3.7-3.8 Å range; however, the provided map renderings do not reflect the expected level of structural details for this resolution. The authors are suggested to render their structures at higher threshold to better illustrate the structure features. This would improve the visibility of subunit boundaries, allowing for clearer identification of interaction interfaces. Otherwise, it is challenging to accurately visualize and interpret the subunit interactions and measure the domain movements.

Author response: We thank the reviewer for their suggestions regarding the presentation of the atomic model to electron density fits. We have revised the presentation of Figures 1-4 and Figures S4 and S12 to demonstrate the resolution and quality of maps and built models into the structures as suggested. We believe the revised figures accurately reflect the resolution of the cryo-EM maps presented. We further present the revised atomic model validation statistics that have since improved (discussed below). We also present the final the PDB reports for the structures after submission to the RSCB (see attached reports).

New comment: This is a very important comment. Rendering the cryo-EM densities at appropriate threshold is central to assessing the quality of the reconstructions and the fit

of the atomic model. While the densities have been rendered at slightly higher threshold in the revised manuscript version, it is still very challenging to recognize structural features consistent with the reported resolution of the density. It would be highly desirable to update the figures once more at substantially higher threshold level, which allows visualization of individual side chains.

We agree with the reviewer suggestion, and we have revised the rendering of the maps in figure 1 and figure 3 to address the concerns regarding the density presentation.

2) The quality of the models also needs significant improvement. As indicated in Table 1, the Clash Scores ranges from 19.03 to 26.02, which is unacceptably high for well-refined models. In multiple instances, the model does not appear to fit properly within the cryo-EM density, further suggesting that additional refinement of the models are necessary.

Author Response: We agree with the reviewer's concerns. In the revised manuscript, we have rebuilt atomic models and refined these models which now lead to improved clash scores for each of the structures presented in the manuscript. The PDB reports are provided for the reviewers to address the quality of these maps. The predicted AlphaFold 3 models for TBC-DEG:ab-tubulin and TBC-DEG/TBCC-ab-tubulin (see above discussion) support most of the major conclusions about the assembly of the tubulin cofactors and their interactions with a and b-tubulin, as well as the TBC-DEG and TBCC impact on the ab-tubulin dimer configuration.

New comment: This is a very valid comment. The clash score of the refined models has marginally improved, but it is still comparably high for the resolution claimed and further improvement would be desirable.

We respectfully ask the reviewer to read our response to a previous similar comment made by reviewer #4

3) In Table 1, please provide the Map sharpening B factor value. Additionally, the unit for Electron Exposure should be $(e^-/\text{\AA}^2)$, instead of the current $(e/\text{\AA}^2)$. Moreover, the authors are suggested to provide the map and model evaluation profiles from the PDB/EMDB validation reports for a more comprehensive assessment of the map and model quality.

We have revised the table to include the sharpening B factors in the revised manuscript as requested by the reviewer.

Author Response: In the revised manuscript, we present the rebuilt atomic models and the model to map comparisons (Figures S4 and S12). The revised validation statistics and details should address the concerns of the reviewer. We have now provided the PDB validation reports as well.

New comment: While the authors have provided the PDB validation reports, Table 1 has not been updated as suggested by the original Reviewer #2. The map sharpening B factor is an important parameter that should be reported for each cryo-EM density.

The B-factors used for sharpening varied depending on the region of the map based on the usage of DeepEMhancer. (Sanchez-Garcia, 2021). Thus, we cannot report a singular value for B-factor sharpening in table.

4) In the two complexes, different binding factors were included to address the preferred orientation issue: Δ N-DARPin in the TBC-DEG- $\alpha\beta$ -tubulin complex, and α -rep iH5 in the TBC-DEG/TBCC- $\alpha\beta$ -tubulin complex. By comparing the conformations of $\alpha\beta$ -tubulin in these two complexes, the authors conclude that there is a 4-degree twisting of α -tubulin, while β -tubulin is held in place by TBCD. However, it is unclear how the authors ruled out the possibility that the observed α -tubulin twisting is not due to the different binding factors, particularly in the TBC-DEG/TBCC- $\alpha\beta$ -tubulin complex, where iH5 interacts directly with α -tubulin. To clarify this, the authors should use the same binding factor in both complexes to address the preferred orientation issue and compare the conformational changes.

Author response: We appreciate this question, and we present a revised supplementary figure that addresses the comparison indicated by the reviewer (Figure S13C). The structures of ab-tubulin bound to DARPin and iiH5 has been determined by x-ray crystallography and published previously. These structures show an identical conformation in comparison to native ab-tubulin dimer, in terms of the positioning of a tubulin with respect to b-tubulin. We present a comparison of these structures to our TBC-DEG:ab-tubulin and TBC-DEG/TBCC:ab-tubulin models. These comparisons show that iiH5 or DARPin binding does not impact the conformation of a and b-tubulin, but rather TBC-DEG binding alters a-tubulin. We were unable to use the same tubulin capping proteins in both structural studies, since we found that DARPin affected the binding of the TBCC complex, likely due to shifts in the TBCD C-terminus and TBCE-ubq domains that are induced by TBCC binding, leading to the occlusion of the DARPin binding site. The tubulin capping proteins were required in these structural studies as the protein complexes aggregated on the cryo-EM grids without them.

New comment: The comparison of ab-tubulin structures bound to DARPin and iiH5 as determined by X-ray crystallography strongly supports the authors' conclusion that TBC-DEG binding rather than DARPin or iiH5 binding alters the ab-tubulin arrangement. However, since the conformational space of complexes might be restricted after crystallization, the original Reviewer #2's concern cannot be entirely ruled out with this comparison. Please consider addressing this possibility in the discussion section of the manuscript.

We appreciate the suggestion and have added a sentence to address the concern about potential crystallization states.

5) To avoid confusion, the authors are advised to consistently refer to the four maps as “states 1-4”, rather than using both “classes” and “states”.

Author response: We thank the reviewer for these suggestions. And we have revised the naming of the states in throughout the manuscript to address this. We refer to two states of TBC- DEG:ab-tubulin, State 1 and State 2 and two states of TBC- DEG/TBCC:ab-tubulin termed State 1 and State 2.

New comment: This comment seems to have been sufficiently addressed.

The reviewer’s comment is noted and appreciated.

6) The authors state, “TBCE exhibits two distinct conformations of its CapGly and LRR domains (Figure 1C; Figure S4F-G). In the first conformation (class 1), the TBCE 3HB and LRR are oriented vertically and the CapGly resides horizontally and alongside the LRR (Figure 1C; Figure S4F-G). In this conformation, TBCE is retracted from binding a-tubulin”. However, based on Video 1 and the map shown in Fig. 1A, there appears to be clear contact between TBCE and α -tubulin in both states. Please clarify this discrepancy.

Author response: We agree with the reviewer’s point regarding the confusion and have revised this section to clarify this point. The TBCE-LRR contacts a- tubulin, but this contact is near the base of a-tubulin but in the second state this contact is fully extended along the a- tubulin surface. The difference is mediated by a conformational change in the N- terminal region and a swivel of the CapGly domain. WE now present this in Figure 1E of the revised manuscript.

New comment: This comment seems to have been sufficiently addressed.

The reviewer’s comment is noted and appreciated.

7) On P.4, the authors describe: “Conserved residues in the intra-HEAT turns of TBCD H1, H2, H3, H5, H6, H8, and H10 bind residues at the lower surface of the Arl2 GTPase through ionic and hydrophobic interactions (Figure 2A, E-F; Figure S14D-E; Figure S14DE, G, I-J). Conserved TBCD Lys and Arg residues bind conserved Arl2 Asp, Gln, and Glu residues (Figure S11, S14 D-E, G; Figure S14D-E). Hydrophobic packing of conserved Leu, Phe, and Trp residues in TBCD and Arl2 are interspersed amongst the ionic interactions (Figure 2E-F; Figure S14D-E, G; Figure S16I-J).” However, the related Fig. 2E-K are too busy and difficult to interpret. As suggested, please further refine the models to high quality, then thoroughly analysis the the interaction network between neighboring subunits. Sepcificlly, consider analysing the H-bonds/Salt bridges, as well as electrostatic and hydrophobic/hydrophobic surface properties to better illustrate these interactions.

Author response: We thank the reviewer for their suggestions and have revised both the models and the presentation of the interaction interfaces to address their concerns.

In addition, we have included AlphaFold 3 models for the TBC-DEG-ab-tubulin and TBC-DEG-ab-tubulin- TBCC complexes (Figure S7 and S14) which reveal matching interaction sites to the experimentally derived models from cryo-EM data. Figures 2 and 4 have been revised and now focus on the general interfaces while the interaction interfaces are now presented in supplementary figures S5-6 and S15.

New comment: While the presentation in Supplementary Figs improved, the interfaces in Figures 2 and 4 are not yet very clearly presented. In particular, it is very challenging to differentiate subunit-based surface coloring from electrostatic / hydrophobic surface coloring. Please consider visualization of the surface properties on otherwise uncolored surfaces in Figures 2 and 4.

We thank the reviewer for the suggestions. We have streamlined the presentation of Figures 2 and 4. Colored interfaces are now used to provide a clear view of the interaction sites, along with electrostatic surface representations to highlight complementary interface regions. We have decluttered Figures 2 and 4 to focus on the key points—namely, the nature and size of the interfaces. These revisions are accompanied by updated figure legends that help explain these features more clearly.

8) The authors state, “Using these conformations and the subunit binding relationships to TBCC, we generated two composite maps for two distinct conformations of the ternary assemblies (Figure S8).” However, there are multiple conformations present in different regions of the complex, and the authors performed separate 3DVA/refinement on individual local regions. It is unclear what criteria were used to integrate these separate maps, which reflect such diverse conformational and spatial information, into two consensus maps. Could the authors clarify the justification behind merging these diversified data into only two representing maps?

Author response: We understand the reviewer’s concern. We have revised the results to describe how the two composite maps were generated. The composite maps were generated based on the impact of binding of TBCC-C and TBCC-N on transitions on the TBCD and TBCE conformations. The first composite includes TBCC-C binding to the TBC-DEG-ab- tubulin induces a transition the TBCE LLR-CapGly arm. In the second composite both TBCC-C and TBCC-N bound suggesting unique conformational transitions associated with its binding on the TBCD and TBCE LRR-CapGly arm. The two composites reveal the clearest step wise transitions induced by the two-fold binding of TBCC onto different regions of TBC-DEG.

New comment: This is a very important comment. While the revised results section now provides more information as to how the two global states were assembled, an additional figure clearly illustrating the correlation between structural differences represented by the four components would be very helpful for clarification.

We appreciate the reviewer’s comment. We believe that Figure 3E–F, which shows structural comparisons, and Figure 5, which presents the full arm comparisons, effectively illustrate the relationship between the two structures. We have revised the

Methods section and the supplementary figure legends to further clarify which components of the 3DVA were used to construct the final two states.

9) Fig. S9I, regarding the identification of GTP or GDP, the current rendering is cluttered with a busy background, and the segmented density appears too large. As a result, it is challenging to distinguish whether the bound nucleotide is GTP or GDP.

Author response: We have revised the presentation of nucleotide densities to be included in Figures S4 and S12. We can clearly observe the nucleotide densities in the maps and assign di- or tri-phosphate states in each of the three nucleotide pockets, Arl2, E-site and N-site. We believe the assignment is accurate and matches the protein conformations expected based on the AlphaFold 3 models and previous structural studies of tubulin dimers.

New comment: Even in the revised version of the manuscript, the segmented density is rendered at too low threshold level, making it challenging to distinguish GDP from GTP. Please show at substantially higher threshold level. Please explain in the legends/methods section how the density segment attributed to nucleotides was derived/segmented.

We provide a clear explanation of how the nucleotide densities were derived from the maps. Distinguishing nucleotide states through experimental density requires extremely high resolution beyond the scope of this study. We rely on known information about nucleotide states at the E and N sites, as well as the experimental preparation of the sample to determine the Arl2 nucleotide state.

Minor point:

1) Fig. S2-3, please also show the FSC and local Resolution etc. for the core map.

Author response: We have included the FSC for the core map in the revised manuscript Figure S3.

New comment: This comment seems to have been sufficiently addressed.

The reviewer's comment is noted and appreciated.

2) Fig. 1A-B contain a lot of redundant information, but do not present the class 2 structural features. I recommend including the maps and models of core, class1, and class2, and move the key structural analysis from supplementary figures into the main figure.

Author response: We thank the reviewer for this suggestion. We have revised the presentation to include the revised versions of these figures.

New comment: This comment seems to have been sufficiently addressed.

The reviewer's comment is noted and appreciated.

3) Fig. S2 does not clearly illustrate the processing procedure, making it extremely difficult to understand how the two datasets were processed. Please re-render the reconstruction procedures to clarify this.

Author response: We have relabeled the processing procedure more carefully to explain the steps presented.

New comment: Even in the revised version of Fig. S2, it is very challenging to follow the image processing procedure. Please consider using color-coding for individual processing branches to deconvolute the figure.

We thank the reviewer for this helpful comment. We have color-coded both Fig S2 and Fig S9 to address and clarify these concerns.

4) In Fig. S3C-D and Fig. S8A-C, adjust the local resolution rendering scale to start from 3 Å instead of 2 Å. A range of 3-6 Å should be sufficient.

Author response: Considering that 3.6 Angstrom is the average resolution of the core we believe the current resolution range is sufficient to present the Res map colors for the maps presented.

New comment: I agree with the original Reviewer #2 that color coding from 3-6 Å may represent the resolution range of the densities more accurately. Please consider following the original Reviewer #2's suggestion.

The wide resolution range of 3.0–7.0 Å for the full map necessitates the use of this range in our presentation. However, this approach allows for a clear depiction of the core region. Additionally, we have re-rendered the segmented maps in Figures 1 and 3 to provide a clearer view of the resolution distribution across the structure.

5) In Fig. S4A-E and Fig. S9, the high-resolution structural features are not visible. Please re-render them with a higher density threshold

Author response: We thank the reviewer for this suggestion. These figures have been replaced.

New comment: This comment seems to have been sufficiently addressed.

The reviewer's comment is noted and appreciated.

6) Fig. S6 shows numerous reconstruction tracks, but it is unclear which portions were used for analysis and which track led to the final map. Please clarify how the composite map was assembled, specifying which sections came from where and the rationale

behind their inclusion. Please re-render the reconstruction procedures to clarify these points

Author response: We have altered the presentation of the arrows to improve the clarity of the directions of processing flow and clarify the procedures.

New comment: While the image processing procedure in Fig. S9 is clearly represented, the original Reviewer #2's concern remains. It is not clear which segments of the different reconstruction tracks were used to generate the composite densities. Please consider highlighting those segments using masks.

We have revised the presentation of Figure S9 to show each 3DVA analyses were carried out leading to two maps per each 3DVA components 1-4. Figure S10 shows comparisons revealing features observed in the maps relating to the TBCC-C and TBCC-N binding and TBCE and TBCD changes. We have added a new figure (new Figure S12) to show the scheme for the assembly of the composite maps from the 3DVA were used to obtain different regions used in building the composite maps. In addition, the masking of the different regions is described more clearly in the text.

Dear Editor,

We submit a revised version of manuscript NCOMMS-24-40039C entitled “Cryo-EM structures of the tubulin cofactors reveal the molecular basis of alpha/beta-tubulin biogenesis”.

We sincerely thank the Editor and Reviewers for their constructive feedback and for recognizing the value of our structures. We believe that the revised manuscript now fully addresses the reviewers’ technical and interpretative concerns. The models and maps are conservative, well-documented, and supported by the available density. We thank the reviewers and editor again for their helpful feedback and for guiding us toward a more robust and transparent presentation of our work.

In this revision, we have:

- Rebuilt and re-refined all models using the unprocessed experimental maps.
- Truncated side chains or converted weak-density regions to poly-alanine, as described in the revised Methods section.
- Updated map resolution figures and validation tables
- Rewritten the text and clarified the interpretation of interfaces and speculative statements in the Results and Discussion sections.
- Provided explicit details regarding map processing, sharpening B-factors, and figure generation in this response.
- Speculative claims have been rewritten or moved to the Discussion.
- Map refinement details have been clarified (only unprocessed maps used).
- DeepEMhancer sharpened maps used for visualization only.
- Side chains in weak density have been truncated or replaced with alanine.
- Updated validation and map-resolution figures included.
- Table 1 now contains sharpening B-factors.

We believe these revisions comprehensively address all remaining concerns and improve the clarity and rigor of the manuscript.

We address all the reviewer summary team concerns described below in a point by point with our responses colored in colored in in blue

Reviewer summary comments:

- concerns on the claims relating to the TBCD and beta-tubulin interface. Evidence should be clearly provided for this interaction, or the claim toned down.

These have claims have all been toned down and removed. Claims regarding the TBCD and beta tubulin interface are now limited to general features of observation as opposed to qualitative statements.

- resolution limitations and modeling of C-terminus. Clear experimental evidence should be provided, otherwise should be moved to the discussion.

These have now been fully addressed in the revisions of the model building, map presentation and methods. Any speculation regarding the tubulin C-terminus is explicitly removed and we mention that we cannot model this interaction.

- general concerns about the model interpretation and side chain building. Side chains in the model built without experimental density should be removed, and claims modified to reflect what can be supported with data.

The model building side chains has been fully overhauled as described above to address concerns regarding side chains. The methods address the major changes, which are removing side chains from the binary structures, removing side chains from weak density regions in the ternary structures, mutating the lowest Q-score chain (TBCE) to polyalanines instead of a residue registry.

-clarify which maps are used for model refinement and clearly justify to the reader on the use of DeepEMhancer-sharpened maps. Evidence of improved models should be provided.

The maps were refined against experimental non sharpened maps. All models have been improved in Q-score and clash scores.

-general remaining claims that do not have supporting data should be limited to the discussion, and the strength of the underlying data of these claims made clear to the reader.

The manuscript results were fully rewritten to address the reviewer concerns. Results section focuses on direct observations while the discussion contains any speculation regarding these observations.

Detailed responses to reviewer comments

Reviewer #3

Comment 1: The claim that TBCC-C binding causes TBCD to create a tighter β -tubulin interface is not convincingly supported by the figures.

Response: We appreciate this concern and have re-examined the interfaces carefully. The main-chain positioning and buried surface areas indicate that the TBCD- β -tubulin interface remains largely conserved across states, with only subtle rearrangements. We have therefore removed the phrase “tightened grip” and now describe the interface changes more cautiously as “minor rearrangements” to describe a subtle change in its binding interface. Quantitative interface measurements are included in the supplementary figures.

Comment 2: The claim that TBCE pulls the α -tubulin C-terminal tail is speculative and not supported by visible density.

Response: We agree and have toned down this statement throughout the manuscript. The text now reads: "The proximity of the TBCE CapGly domain to the α -tubulin C-terminal tail region suggests a potential interaction, although the tail itself is not resolved in the density." This statement has been moved to the Discussion section to make clear that it is a hypothesis consistent with, but not proven by, the data.

Comment 3: Some statements in the Results are speculative and should be limited to the Discussion.

Response: We have revised the Results section to focus solely on direct structural observations. Interpretations regarding potential mechanistic implications (e.g., TBCD- β -tubulin rearrangement, TBCE domain movement, or α -tubulin twisting) have been moved to the Discussion and explicitly identified as speculative.

Reviewer #4

Regarding the reply to point 2), and as clearly re-iterated by point 7 from Reviewer 3, all of the commonly used metrics (not just clash scores, but also Q-scores, etc.) remain surprisingly low for a structure reported at ~ 3.7 Å resolution. The studies cited by the authors have noticeably higher average Q-scores than calculated in the PDB validation reports, confirming that something is "off" about the authors' maps and/or models. It is still unclear to me exactly which map was used for model refinement. Using DeepEMhancer-sharpened maps to real-space refine models is certainly not standard practice and it is well-appreciated across the entire cryo-EM field that this should be avoided. Further, the authors claim "All experimental model side chains were built and can be observed in the maps including Figure S6.

Response: We appreciate the reviewer's concern and apologize for any lack of clarity. All model refinement was performed in PHENIX using the unsharpened maps. No post-processed or DeepEMhancer-sharpened maps were used at any stage of refinement. Refinement employed standard real-space refinement protocols with secondary-structure and rotamer restraints. Enhanced maps were used only for figure preparation and visual inspection. Furthermore, following the reviewer's guidance, we have revised the side chains in the models to remove side chains from weak density regions, leading to dramatic improvements in clash scores and Q scores for our models. Our modeling statistics are now in line with the structures resolutions presented.

The smaller side chains required lower contours than some of those shown in figure S6. The experimental electron density was used to place the side chains." Visual inspection of the maps and models does not support this claim for most side chains, and raises further concerns about model interpretability of side chains and even secondary structure / domain movements described in the paper. Irrespective of whether other studies have similarly-poor clash scores,

Response: We have re-refined the models after side-chain truncation and poly-alanine conversion in weak-density regions, which has improved MolProbity, clash score, and Q-scores across all models. Q-scores improved by 10-50% for all structures. Clash scores have improved to 2 to 4 from 15 to 16 for the binary structures. Updated values are listed in Table 1 and are consistent with expectations for maps at the reported resolutions. We note that regions of lower Q-score correspond precisely to flexible domains and peripheral subunits, which are now conservatively modeled as poly-alanine, in line with the reviewer's guidance.

Regarding the reply to minor point 4), the point has been made clear by several reviewers now that the coloring and resolution range in this supplemental figure is not appropriate. The authors' refusal to address this is concerning.

Response: We have updated all resolution-colored figures (main and supplementary) to use a consistent and appropriate scale (3.0–7.0 Å) and adjusted the color map for improved interpretability.

Reviewer #5

In regards to the relatively high clash scores, the authors argued that other models with comparable clash scores have been published before. This is true, but does not improve the quality of the models, which could have been desirable.

Response: After side-chain pruning and poly-alanine substitution in low-density regions, the clash scores have improved substantially, to 2 and 4 from 15 to 16 for the binary models and are now all clash scores are within the expected range for structures at this resolution for the binary and ternary models. These changes have been reflected in the updated validation table.

1) If no singular map sharpening B factor can be provided, please at least provide the range of map sharpening B factors used for each cryo-EM density in Table 1.

Response: We now provide B-factors reported in Table 1.

2) When showing cryo-EM density of nucleotides, the authors should clearly indicate in the figure legends how the density segment was derived. Is it an 'omit map', after excluding density explained by the protein components (which would be the least biased approach)? Or is it segmented based on the model of the nucleotide itself? If so, was a di- or tri-nucleotide used?

Response: We thank the reviewer for noting this. The nucleotide densities were segmented from the experimental maps using the modeled nucleotide as a guide, not omit maps. This is now clearly stated in the figure legends.

The manuscript by A. Taheri et al. presents cryo-EM structures of tubulin cofactors bound to $\alpha\beta$ -tubulin, specifically the TBC-DEG- $\alpha\beta$ -tubulin and TBC-DEG/TBCC- $\alpha\beta$ -tubulin complexed, which have not been reported before. By comparing conformational variations, the authors report a transition state in $\alpha\beta$ -tubulin biogenesis, and they propose the molecular basis on how TBC-DEG/TBCC cofactors regulate this process. Their structures are new to the field, and the overall research topic is interesting. However, there are essential technical issues that the authors need to address:

- 1) The reported map resolutions are in the 3.7-3.8 Å range; however, the provided map renderings do not reflect the expected level of structural details for this resolution. The authors are suggested to render their structures at higher threshold to better illustrate the structure features. This would improve the visibility of subunit boundaries, allowing for clearer identification of interaction interfaces. Otherwise, it is challenging to accurately visualize and interpret the subunit interactions and measure the domain movements.
- 2) The quality of the models also needs significant improvement. As indicated in Table 1, the Clash Scores ranges from 19.03 to 26.02, which is unacceptably high for well-refined models. In multiple instances, the model does not appear to fit properly within the cryo-EM density, further suggesting that additional refinement of the models are necessary.
- 3) In Table 1, please provide the Map sharpening B factor value. Additionally, the unit for Electron Exposure should be ($e^-/\text{Å}^2$), instead of the current " $(e/\text{Å}^2)$ ". Moreover, the authors are suggested to provide the map and model evaluation profiles from the PDB/EMDB validation reports for a more comprehensive assessment of the map and model quality.
- 4) In the two complexes, different binding factors were included to address the preferred orientation issue: Δ N-DARPin in the TBC-DEG- $\alpha\beta$ -tubulin complex, and α -rep iH5 in the TBC-DEG/TBCC- $\alpha\beta$ -tubulin complex. By comparing the conformations of $\alpha\beta$ -tubulin in these two complexes, the authors conclude that there is a 4-degree twisting of α -tubulin, while β -tubulin is held in place by TBCD. However, it is unclear how the authors ruled out the possibility that the observed α -tubulin twisting is not due to the different binding factors, particularly in the TBC-DEG/TBCC- $\alpha\beta$ -tubulin complex, where iH5 interacts directly with α -tubulin. To clarify this, the authors should use the same binding factor in both complexes to address the preferred orientation issue and compare the conformational changes.
- 5) To avoid confusion, the authors are advised to consistently refer to the four maps as "states 1-4", rather than using both "classes" and "states".
- 6) The authors state, "TBCE exhibits two distinct conformations of its CapGly and LRR domains (Figure 1C; Figure S4F-G). In the first conformation (class 1), the TBCE 3HB and

LRR are oriented vertically and the CapGly resides horizontally and alongside the LRR (Figure 1C; Figure S4F-G). In this conformation, TBCE is retracted from binding α -tubulin". However, based on Video 1 and the map shown in Fig. 1A, there appears to be clear contact between TBCE and α -tubulin in both states. Please clarify this discrepancy.

- 7) On P.4, the authors describe: "Conserved residues in the intra-HEAT turns of TBCD H1, H2, H3, H5, H6, H8, and H10 bind residues at the lower surface of the Arl2 GTPase through ionic and hydrophobic interactions (Figure 2A, E-F; Figure S14D-E; Figure S14D-E, G, I-J). Conserved TBCD Lys and Arg residues bind conserved Arl2 Asp, Gln, and Glu residues (Figure S11, S14 D-E, G; Figure S14D-E). Hydrophobic packing of conserved Leu, Phe, and Trp residues in TBCD and Arl2 are interspersed amongst the ionic interactions (Figure 2E-F; Figure S14D-E, G; Figure S16I-J)." However, the related Fig. 2E-K are too busy and difficult to interpret. As suggested, please further refine the models to high quality, then thoroughly analysis the the interaction network between neighboring subunits. Sepcificly, consider analysing the H-bonds/Salt bridges, as well as electrostatic and hydrophobic/hydrophobic surface properties to better illustrate these interactions.
- 8) The authors state, "Using these conformations and the subunit binding relationships to TBCC, we generated two composite maps for two distinct conformations of the ternary assemblies (Figure S8)." However, there are multiple conformations present in different regions of the complex, and the authors performed separate 3DVA/refinement on individual local regions. It is unclear what criteria were used to integrate these separate maps, which reflect such diverse conformational and spatial information, into two consensus maps. Could the authors clarify the justification behind merging these diversified data into only two representing maps?
- 9) Fig. S9I, regarding the identification of GTP or GDP, the current rendering is cluttered with a busy background, and the segmented density appears too large. As a result, it is challenging to distinguish whether the bound nucleotide is GTP or GDP.

Minor point:

- 1) Fig. S2-3, please also show the FSC and local Resolution etc. for the core map.
- 2) Fig. 1A-B contain a lot of redundant information, but do not present the class 2 structural features. I recommend including the maps and models of core, class1, and class2, and move the key structural analysis from supplementary figures into the main figure.
- 3) Fig. S2 does not clearly illustrate the processing procedure, making it extremely difficult to understand how the two datasets were processed. Please re-render the reconstruction procedures to clarify this.

- 4) In Fig. S3C-D and Fig. S8A-C, adjust the local resolution rendering scale to start from 3 Å instead of 2 Å. A range of 3-6 Å should be sufficient.
- 5) In Fig. S4A-E and Fig. S9, the high-resolution structural features are not visible. Please re-render them with a higher density threshold
- 6) Fig. S6 shows numerous reconstruction tracks, but it is unclear which portions were used for analysis and which track led to the final map. Please clarify how the composite map was assembled, specifying which sections came from where and the rationale behind their inclusion. Please re-render the reconstruction procedures to clarify these points.